# Molecular response to the non-lytic peptide bac7 (1–35) triggers disruption of *Klebsiella pneumoniae* biofilm

Robert L. Beckman I.V.[1], Berta Martinez[1], Flor Z. Santiago[1], Gabriela N. Echeverria[1], Bruno V. Pinheiro[1], Marcelo D.T. Torres[2,3,4,5], Logan Suits[6], Shantal Garcia[7], Paeton L. Wantuch[7], Cesar de la Fuente-Nunez[2,3,4,5], Prahathees Eswara[6], David A. Rosen[7], Renee M. Fleeman[1]*

1 Burnett School of Biomedical Sciences, College of Medicine, University of Central Florida, Orlando, Florida, United States of America, 2 Machine Biology Group, Departments of Psychiatry and Microbiology, Institute for Biomedical Informatics, Institute for Translational Medicine and Therapeutics, Perelman School of Medicine, University of Pennsylvania, Philadelphia, Pennsylvania, United States of America, 3 Departments of Bioengineering and Chemical and Biomolecular Engineering, School of Engineering and Applied Science, University of Pennsylvania, Philadelphia, Pennsylvania, United States of America, 4 Department of Chemistry, School of Arts and Sciences, University of Pennsylvania, Philadelphia, Pennsylvania, United States of America, 5 Penn Institute for Computational Science, University of Pennsylvania, Philadelphia, Pennsylvania, United States of America, 6 Department of Molecular Biosciences, University of South Florida, Tampa, Florida, United States of America, 7 Department of Pediatrics, Washington University School of Medicine, St. Louis, Missouri, United States of America

* Renee.Fleeman@ucf.edu

## Abstract

*Klebsiella pneumoniae* is becoming increasingly difficult to treat as multidrug-resistant (MDR) strains become more prevalent. The formation of biofilm heightens this threat by embedding bacterial cells in a polysaccharide-rich matrix that limits antibiotic penetration. Here we dissect the anti-biofilm bovine host-defense cathelicidin peptide fragment bac7 (1–35), exploring its anti-biofilm mechanism, evaluating its ability to curb colonization of the vital organs by hypervirulent *K. pneumoniae*, and testing its breadth of activity against diverse clinical isolates. Transcriptomic profiling revealed that bac7 (1–35) simultaneously compromises the bacterial membrane and inhibits ribosomal function, a dual assault that precipitates rapid biofilm collapse and blocks bacterial spread. Further, bac7 (1–35) eradicated the strongest biofilms produced by MDR clinical isolates in the Multidrug-Resistant Organism Repository and Surveillance Network (MRSN) diversity panel. Although bac7 (1–35) kills bacterial cells via a cytosolic mechanism, membrane interaction profiles varied among MRSN isolates, correlating with differential peptide translocation. In a delayed-treatment murine skin-abscess model, bac7 (1–35) halted *in vivo* colonization of the vital organs by the hypervirulent strain NTUH-K2044. Collectively, these results delineate a multifaceted mode of action for bac7 (1–35) and underscore its therapeutic promise against biofilm-associated MDR *K. pneumoniae* infections.

which permits unrestricted use, distribution, and reproduction in any medium, provided the original author and source are credited.

**Data availability statement:** All relevant data are within the manuscript and its Supporting Information files.

**Funding:** This work was supported by National Institute of Allergy and Infectious Diseases of the National Institutes of Health under award number R00AI163295 awarded to R.M.F. C.F.-N. holds a Presidential Professorship at the University of Pennsylvania and acknowledges funding from the Procter & Gamble Company, United Therapeutics, a BBRF Young Investigator Grant, the Nemirovsky Prize, Penn Health-Tech Accelerator Award, Defense Threat Reduction Agency grants HDTRA11810041 and HDTRA1-23-1-0001, and the Dean's Innovation Fund from the Perelman School of Medicine at the University of Pennsylvania. Research reported in this publication was supported by the Langer Prize (AIChE Foundation), the NIH R35GM138201, and DTRA HDTRA1-21-1-0014. The funders had no role in study design, data collection and analysis, decision to publish, or preparation of the manuscript.

**Competing interests:** I have read the journal's policy and the authors of this manuscript have the following competing interests: CFN is a co-founder and scientific advisor to Peptaris, Inc., provides consulting services to Invaio Sciences and is a member of the Scientific Advisory Boards of Nowture S.L., Peptidus, and Phare Bio. CFN is also on the Advisory Board of the Peptide Drug Hunting Consortium (PDHC). The de la Fuente Lab has received research funding or in-kind donations from United Therapeutics, Strata Manufacturing PJSC, and Procter & Gamble, none of which were used in support of this work. MDTT is a co-founder and scientific advisor to Peptaris, Inc. The remaining authors declare no competing interests.

## Author summary

*Klebsiella pneumoniae* is a top-priority pathogen for new therapies, with many strains already approaching pan-drug resistant status. Biofilm formation further complicates treatment, yet biofilm-active therapeutics have not reached the clinic, in part because we still lack a detailed understanding of how to disrupt these impenetrable structures. Antimicrobial peptides are promising candidates and have shown biofilm-disruption potential. Here we provide mechanistic insight into how a host defense peptide dismantles pre-formed *K. pneumoniae* biofilms. We find that the peptide's dual targeting of bacterial membranes and ribosomes triggers dispersal from the biofilm state and concomitantly downregulates factors required for surface attachment and extracellular matrix production. This mechanism involves a protein that, to our knowledge, has not been characterized in *K. pneumoniae*. Our findings reveal a switch that can be leveraged to reprogram biofilm maintenance toward dispersal in *K. pneumoniae*, advancing the path to peptide-based antibiofilm therapeutics.

## Introduction

*Klebsiella pneumoniae* has emerged as a priority pathogen in recent years due to its extreme drug resistance and the emergence of virulence traits [1–8]. Over the past few decades, two distinct pathotypes have been reported: classical *K. pneumoniae* (cKp), many of which are MDR and commonly infect patients with underlying chronic diseases; and hypervirulent *K. pneumoniae* (hvKp), which can afflict healthy individuals in the community [5–7]. Regardless of pathotype, biofilm formation is associated with 60–80% of hospital-associated infections and increases the antibiotic resistance and immune evasion of isolates that may otherwise be susceptible [9]. This complex matrix is often impenetrable to standard antibiotics and protects the embedded bacterial cells from the immune system, resulting in recalcitrant hospital-associated infections [9]. Despite the significance of these infections, effective strategies to penetrate and disrupt biofilms remain an unmet need.

Biofilm formation is essential for bacterial survival under environmental stress and is a highly regulated process involving transcriptional changes guided by environmental stressors and signal responses [10,11]. *K. pneumoniae* infection related biofilms complicate treatment of wound infections, catheter infections, and ventilator associated pneumonia infections [12]. *K. pneumoniae* biofilm formation has been shown to be attributed to lipopolysaccharide (LPS) and polysaccharide production, expression of fimbriae, and iron metabolism [12]. However, *K. pneumoniae* has an extensive accessory genome leading to a large species diversity, which complicates studies, as the isolate and the conditions tested influence the factors important for biofilm formation [13]. For example, type I and type III fimbriae have both been shown to be c-di-GMP regulated and play an important role in attachment during the initial phase of biofilm but their respective impact varies based on the conditions

and strains tested [12,14,15]. Similarly, the role of capsular polysaccharides has been shown to have a positive or negative impact on biofilm formation depending on the isolate tested [16,17]. However, other extracellular polysaccharides important for biofilm formation such as poly-N-acetylglucosamine (PNAG) and cellulose have not been characterized as well as capsular polysaccharides in *K. pneumoniae* [18,19]. Cellulose is an aggregative, secreted polysaccharide that is unique from the immune protective, attached capsular polysaccharide [20]. Importantly, cellulose has been highlighted in *K. pneumoniae* studies to be an important factor in biofilm formation [19,21]. During infection, environmental stimuli, such as limited nutrients, temperature, and pH changes, promote biofilm formation and persistent infections [11,22]. Signaling molecules that respond to environmental stressors act as molecular switches to coordinate the transcriptional changes necessary for biofilm formation [22,23]. Fimbriae and cellulose have been shown to be important for biofilm formation and regulated by the second messenger cyclic diguanosine monophosphate (c-di-GMP) [24–27]. Due to its important role in biofilm formation, c-di-GMP has been viewed as a promising drug target for combating biofilm formation, which can be targeted directly or indirectly by interfering with cellular processes involved in its regulation [22]. However, targeting c-di-GMP directly may prove challenging as there are multiple diguanylate cyclase enzymes with redundant abilities to produce c-di-GMP [28].

Antimicrobial peptides have shown promise as therapeutics targeting MDR Gram-negative pathogens and have displayed the potential to disrupt biofilms [29–31]. Bactenecin 7 is a proline-rich cathelicidin peptide derived from bovine intestinal neutrophils, where the 60 amino acid pro-peptide is stored until the mammalian elastase enzyme cleaves the pro-peptide to produce the active peptide fragment bac7 (1–35) [32]. Bac7 (1–35) belongs to a class of proline-arginine rich peptides that are unique host-defense peptides, as they exert intracellular antimicrobial effects without causing membrane lysis like most antimicrobial peptides [33,34]. Instead, their primary mode of action is the inhibition of protein translation through binding the 50S ribosomal subunit [35–37]. Entry into the cytoplasm is typically mediated by stereospecific binding to the SbmA transporter allowing for peptide translocation without membrane lysis [37]. However, the mechanism of action of bac7 (1–35), henceforth referred to as bac7, varies between bacterial genera, with membrane-targeting activity observed in *Pseudomonas aeruginosa* and intracellular inhibition documented in *Salmonella enterica* and *Escherichia coli* [36]. Although it has not been determined exactly why bac7 targets the membrane of *P. aeruginosa* yet targets the ribosome of *S. enterica* and *E. coli*, we hypothesize the membrane composition differences play a major role. Furthermore, the exact mechanism of bac7 killing of *K. pneumoniae* has not been thoroughly studied to compare to these other bacterial genera. Our previous work has shown that bac7 binds and aggregates with extracellular polysaccharides to allow for the collapse of hvKp NTUH-K2044 biofilms [38]. Importantly, the decreased mucoidy in treated biofilm supernatant highlights the potential of bac7 to decrease the hypermucoviscous nature of hvKp and suggests it might prevent *in vivo* colonization of the vital organs by hvKp. However, the contribution of bac7's membrane interactions to its mechanism of action against *K. pneumoniae* remains unclear.

Protein synthesis inhibition has been shown to induce toxic levels of adenosine triphosphate (ATP) and membrane stress [39,40]. Elevated ATP is dangerous for bacteria because the abundance of ATP sequesters the cytosolic magnesium leading to magnesium starvation and membrane instability [39,41,42]. Therefore, it is understandable that magnesium transport and membrane lipid modifications are intimately connected through the PhoPQ two-component global regulatory system, which is activated by membrane stress and low magnesium conditions [43]. Within the PhoP regulon, the protein MgtC has been shown to not only be activated by the response regulator PhoP, but also to self-regulate through stem-loop formation in the *mgtCBR* leader mRNA. This region contains two short open reading frames, where the presence of elevated ATP increases *mgtC* expression [44]. Bruna et al, has recently described that the function of MgtC in *S. enterica* is to inhibit inorganic phosphate (Pi) uptake [39]. Inhibition of Pi uptake by MgtC decreases the production of new ATP from ADP and works coordinately with MgtA, a transporter that brings extracellular magnesium into the cell. In *S. enterica*, MgtC activity has been associated with a decrease in levels of c-di-GMP, although the exact mechanism has yet to be defined [45]. However, decreased intracellular Pi has been shown to interfere with many energetic processes

including oxidative phosphorylation, tricarboxylic acid cycle, and c-di-GMP production [46–48]. Furthermore, the role of MgtC has been shown to vary depending on the lifestyle of the bacteria, as it is important for intracellular survival in *S. enterica,* yet impacts biofilm formation in primarily extracellular pathogens [49,50]. Although MgtC has been studied in multiple bacterial genera, the role of this protein has yet to be studied in *K. pneumoniae*.

Here, we demonstrate that bac7 induces both membrane stress and ribosome inhibition, embodying a dual mechanism of action against *K. pneumoniae*. We reveal bac7 induces a membrane stress response without cell lysis, and activation of *mgtC*-mediated biofilm disruption by modulating c-di-GMP activated cellulose and type I fimbriae (Fig 1). Expanding our analysis using the Multidrug-Resistant Organism Repository and Surveillance Network (MRSN) *K. pneumoniae* diversity panel [15], we found variable bac7 induced membrane depolarization amongst diverse isolates and heterogeneous peptide uptake within single populations. Testing bac7 using an *in vivo* skin abscess model revealed bac7 treatment has the potential to decrease colonization of the vital organs by hvKp NTUH-K2044. Our results show that the dual mechanism of action of bac7 towards *K. pneumoniae* triggers a molecular response to decrease cellulose and fimbriae leading to biofilm disruption and topical treatment of hvKp NTUH-K2044 with bac7 decreased *in vivo* colonization of the vital organs in a skin abscess infection.

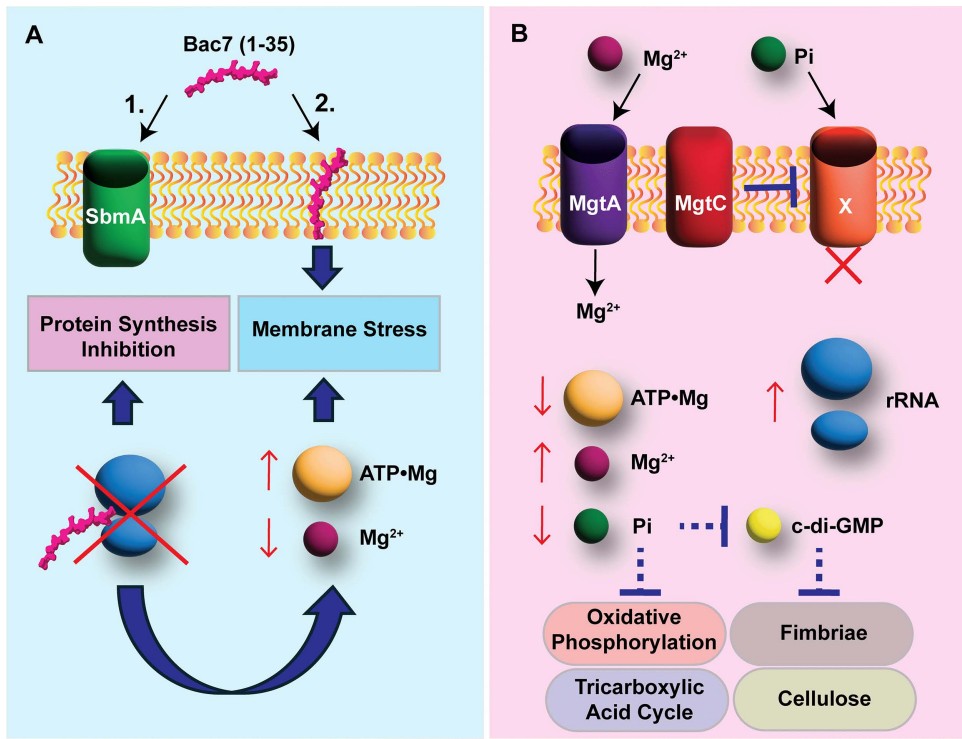

**Fig 1. Bac7 dual killing mechanism and *K. pneumoniae* molecular response to elevated ATP.** The figure shows the mechanisms of *K. pneumoniae* entry and killing by bac7 next to the bacterial response to combat bac7 dual membrane and ribosomal stress. A shows the two mechanisms of cell entry by bac7 where black arrows indicate 1) transporter mediated uptake and 2) membrane translocation into the cytosol. Red arrows indicate the metabolite changes upon bac7 treatment and blue directional arrows show the causative relationship towards the dual killing mechanisms. B shows that in response to bac7, *K. pneumoniae* activates MgtA to transport extracellular magnesium ions into the cell (black arrows), and MgtC to inhibit uptake of phosphate ions (Pi) through a transporter yet to be defined in *S. enterica*. These actions decrease intracellular Pi and ATP, which consequently increases transcription of rRNA and intracellular magnesium concentrations (red arrows). Pi limitation may be the connection to the decreases in oxidative phosphorylation, the tricarboxylic acid cycle, and c-di-GMP regulated fimbriae and cellulose operons in our transcriptional profiling (dashed blunt arrow).

## Results

### Transcriptional insight reveals complex mechanism of killing and biofilm disruption

There has been a diversity of reported mechanisms of bacterial killing by bac7, and these mechanisms have been shown to vary between bacterial genera [36,51]. Given the multiple potential mechanisms, we aimed to identify the mechanism of killing in hvKp and to understand the rapid biofilm collapse that occurs before viability changes observed in our original study [38]. We hypothesized that there is a molecular mechanism of peptide-mediated biofilm disruption. We therefore investigated the molecular response of hvKp to bac7 using RNA sequencing of hvKp NTUH-K2044 with 7.5 µmol L$^{-1}$ of bac7 for 30 minutes, which was the concentration that we have previously shown to cause rapid disruption of NTUH-K2044 biofilms [38,52].

Due to the concentration of peptide that disrupts biofilm being well above the minimal inhibitory concentration (MIC), there were a large number of genes that displayed significant differential expression (2,511 genes -log10(FDR) > 2) following treatment with bac7 compared to the no treatment control group, with 1,202 genes upregulated (log2 Fold Change > 1) and 1,309 genes downregulated (log2 Fold Change < -1) (S1 Data). To provide context to the overall gene expression changes with bac7, we used KEGG and STRING database analyses [53–55]. The KEGG pathway enrichment analysis revealed ribosome (54 genes upregulated, p-value 1.5e$^{-21}$), oxidative phosphorylation (37 genes downregulated, p-value 7.5e$^{-10}$), and the TCA cycle (28 genes downregulated, p-value 1.6e$^{-8}$) pathways were the most changed with treatment (S1 Fig and S1 Table). When considering gene ontology, we found genes associated with energy production were again the most enriched (S2 Fig). These pathways have been shown to be common metabolic responses to antibiotic treatment but also antimicrobial peptides [56,57]. The STRING protein-protein interaction network analysis (FDR < 1 e$^{-10}$, Log2 FC > 1.5) revealed significantly more interactions than expected indicating a potential biological connection between the differentially expressed genes (S2 Data). We found two large clusters of interacting proteins corresponding to protein translation (increased expression) and oxidative phosphorylation (decreased expression), with F$_o$F1 ATP synthase subunit epsilon (KP1_RS25675) serving as the connecting node between these groups (S3 Fig). Interestingly, we found the ribosomal cluster connected with transcription factors to reveal a footprint of membrane stress. Specifically, *rpoB* and *rpoC* genes encoding for RNA polymerase subunits were decreased in expression with bac7 indicating potential overall decrease in transcription levels (Table 1). Conversely, we found *rpoE*, a gene encoding for a membrane stress response transcription factor was increased in expression, suggesting membrane stress is induced by bac7. Looking downstream at RpoE regulated genes, we found *ompC* to have decreased expression, possibly in response to the increased expression of *rseA*, encoding for an anti-sigma factor that blocks RpoE activity (Table 1). Our data suggest that the bac7 mechanism of killing *K. pneumoniae* involves both ribosome inhibition and membrane stress.

We used False Discovery Rate (FDR) to decrease the number of false positives (FDR < 1 e$^{-10}$) when looking at the individual genes displaying the most significant changes in expression following treatment with bac7. With this strict cutoff, we found changes not only indicating membrane and ribosomal stress but also changes suggesting a switch out of biofilm formation state (Fig 2A). Specifically, polyamines have been shown to be important for biofilm formation [58] and are coordinately regulated with magnesium in *S. enterica* [58]. We found lysine decarboxylase (*cadA*), important for polyamine cadaverine production, displayed the greatest decrease in expression of all genes in our profile (Table 1) [59]. We also saw decreased expression of the genes involved in export of the polyamine putrescine. Another biofilm related transcriptional change we found was increased expression of toxin-antitoxin system genes, shown to be involved in both resistance to antimicrobial peptides and biofilm formation [56]. Finally, we found significantly increased expression of *mgtC*, that encodes a protein shown to inhibit Pi uptake [39]. Our analysis revealed increased expression of phosphate transporter genes *phoB, phoR*, *ptsC*, and *ptsS* that aligns with heterologous *mgtC* expression and phosphate limitation in *S. enterica* (Table 1) [39]. With the major role of Pi in cellular energetics, we hypothesize MgtC-mediated Pi depletion may be key to the changes to cellular metabolism genes we found in our KEGG pathway analysis (S1 Fig). Intriguingly, we

**Table 1. Expression changes with bac7 treatment.**

| Genes categorized by function | Log2 FC | pValue | FDR |
|---|---|---|---|
| **Housekeeping RNAP subunits** | | | |
| rpoB | -2.5 | 1.9E-12 | 5.0E-11 |
| rpoC | -4.1 | 1.2E-14 | 1.5E-12 |
| **Membrane stress transcription factor and regulon** | | | |
| rpoE | +3.2 | 5.2E-13 | 3.8E-11 |
| rseA | +3.0 | 2.4E-13 | 1.2E-11 |
| ompC | -4.0 | 6.0E-15 | 1.2E-12 |
| **Polyamine export genes** | | | |
| cadA | -9.3 | 5.5E-16 | 4.6E-13 |
| cadB | -2.9 | 2.5E-11 | 3.3E-10 |
| sapC | -5.7 | 1.6E-13 | 9.5E-12 |
| sapD | -4.5 | 6.5E-13 | 2.3E-11 |
| sapF | -6.3 | 1.5E-13 | 9.1E-12 |
| **Toxin/Antitoxin system** | | | |
| vapB | +4.7 | 2.5E-12 | 6.0E-11 |
| vapC | +5.1 | 2.4E-07 | 8.0E-07 |
| **PhoP regulon** | | | |
| phoP | +1.3 | 1.6E-08 | 7.2E-08 |
| mgtC | +5.4 | 3.6E-16 | 4.6E-13 |
| **Phosphate uptake** | | | |
| phoB | +2.1 | 6.9E-09 | 3.6E-08 |
| phoR | +1.5 | 2.4E-08 | 1.0E-07 |
| ptsC | +1.1 | 2.7E-06 | 7.0E-06 |
| ptsS | +2.9 | 2.7E-12 | 6.3E-11 |
| **F$_o$F1 ATP synthase subunits** | | | |
| atpA | -4.6 | 3.0E-14 | 2.9E-12 |
| atpC | -5.6 | 2.3E-15 | 8.7E-13 |
| atpD | -5.9 | 1.4E-15 | 7.0E-13 |
| atpG | -5.4 | 7.9E-15 | 1.4E-12 |
| **ATP-binding cassette domain-containing protein** | | | |
| KP1_RS24555 | +6.8 | 4.4E-17 | 2.1E-13 |
| **Type I fimbriae operon** | | | |
| fimG | -7.6 | 4.9E-14 | 4.6E-13 |
| fimH | -7.4 | 5.1E-15 | 1.1E-12 |
| fimK | -7.0 | 2.8E-15 | 9.4E-13 |
| fimC | -6.7 | 4.0E-13 | 1.7E-11 |
| fimD | -4.9 | 6.8E-15 | 1.3E-12 |
| fimA | -1.3 | 2.5E-09 | 1.5E-08 |
| **Cellulose operon** | | | |
| bcsB | -6.6 | 3.6E-14 | 2.3E-12 |
| bcsZ | -6.3 | 2.5E-14 | 2.5E-12 |
| bcsC | -5.5 | 2.4E-13 | 1.2E-11 |
| bcsG | -3.9 | 7.7E-12 | 1.4E-10 |
| bcsA | -3.2 | 5.6E-12 | 1.2E-10 |
| bcsB2 | -3.0 | 4.5E-11 | 5.3E-10 |
| bcsA2 | -2.9 | 3.3E-12 | 7.6E-11 |

*(Continued)*

**Table 1.** (Continued)

| Genes categorized by function | Log2 FC | pValue | FDR |
|---|---|---|---|
| *bcsF* | -2.6 | 6.5E-07 | 2.0E-06 |

found bac7 treatment significantly decreased expression of $F_OF1$ ATP synthase genes, which was found as a major connecting node in our STRING analysis (S3 Fig). Furthermore, the most significantly upregulated gene in our analysis (KP1_RS24555) encodes for an ATP-binding cassette domain-containing protein that when blasted on NCBI shows 99% protein homology to EttA (GenBank: CAH5943538.1), that acts as an energy-dependent translational throttle shown to respond to cellular changes in ATP [60]. In line with our hypothesis that Pi depletion by MgtC decreases c-di-GMP levels, we found the c-di-GMP activated type 1 fimbriae and cellulose operons [26,45] (*fimACDGHK, bcsA1B1CFGZ, and bcsA2B2*) had decreased expression with bac7 treatment (Table 1).

We hypothesized the decreased expression of cellulose and fimbriae operon genes are due to the increased expression of the gene encoding for the MgtC family protein. There are no studies of this protein in *K. pneumoniae*; however in *S. enterica,* expression of *mgtC* is controlled by the global response regulator PhoP in response to membrane stress [45,61], and we observed increased expression of *phoP* in our analysis (Table 1). Once transcribed, hairpin loops in the leader mRNA increase expression of the *mgtCBR* operon in response to elevated ATP under stressed conditions as seen with ribosomal binding by chloramphenicol [44,62]. MgtC relieves ATP toxicity by reducing cytoplasmic Pi to decrease production of new ATP, which frees the magnesium ions to rescue the magnesium starvation induced by ribosome inhibition [39,44]. Although the exact mechanism has not been defined, activation of MgtC in *S. enterica* also results in a reduction of c-di-GMP, which is an allosteric activator of cellulose and fimbriae expression (type 1 and 3) [26,45]. We used RT-qPCR to validate the bac7 induced expression changes in *phoP* and *mgtC*, as well as the downstream *bcsA* and *fimH*, known to be transcriptionally activated by c-di-GMP [45]. Using the same bac7 treatment conditions that were used for RNA sequencing revealed significantly increased expression of both *phoP* and *mgtC* when compared to the no treatment control (Fig 2B). When looking at c-di-GMP allosterically activated genes important for biofilm shown in our sequencing, bac7 treatment significantly decreased the expression of *bcsA* and *fimH* compared to the no treatment control (Fig 2C).

To understand the metabolic changes associated with bac7 treatment we quantified both ATP and c-di-GMP using the BacTiter-Glo microbial cell viability kit and Cyclic di-GMP ELISA kit, respectively. Following 30-minutes treatment of hvKp NTUH-K2044, we found elevated ATP with bac7 treatment (Fig 2D). Control experiments using chloramphenicol (ribosomal binding control) and polymyxin B (lytic peptide control) treated samples demonstrated the expected increase and decreased in ATP, respectively (S4A Fig). However, when comparing bac7 treatment to chloramphenicol, a small molecule protein synthesis inhibitor, we found bac7 treatment resulted in sustained elevated levels of ATP and increased *mgtC* expression (S4B Fig). Conversely, we found significantly decreased c-di-GMP with bac7 treatment (Fig 2E). Control experiments revealed a similar decrease in c-di-GMP with chloramphenicol treatment, but not with polymyxin B (S4C Fig). To assess if our transcriptional changes resulted in a phenotypical change, we quantified the abundance of FimA produced by hvKp NTUH-K2044 following treatment with increasing concentrations of bac7 compared to no treatment using FimA immunoblot analysis (S4D Fig). We found decreased FimA by immunoblot following treatment with 7.5 µmol L$^{-1}$ bac7 (Fig 2F) and saw a significant reduction when quantified across several independent experiments (P = 0.0067) (Fig 2G). Further, we performed transmission electron microscopy (TEM) and observed more bacterial cells expressing fimbriae in the control group compared to bacterial cells treated with 7.5 µmol L$^{-1}$ bac7 (Fig 2H and 2I). Together our transcriptional analysis indicated bac7 causes both cytosolic and membrane stress, and we propose this dual stress causes an *mgtC*-mediated biofilm disruption by decreasing c-di-GMP and production of type I fimbriae.

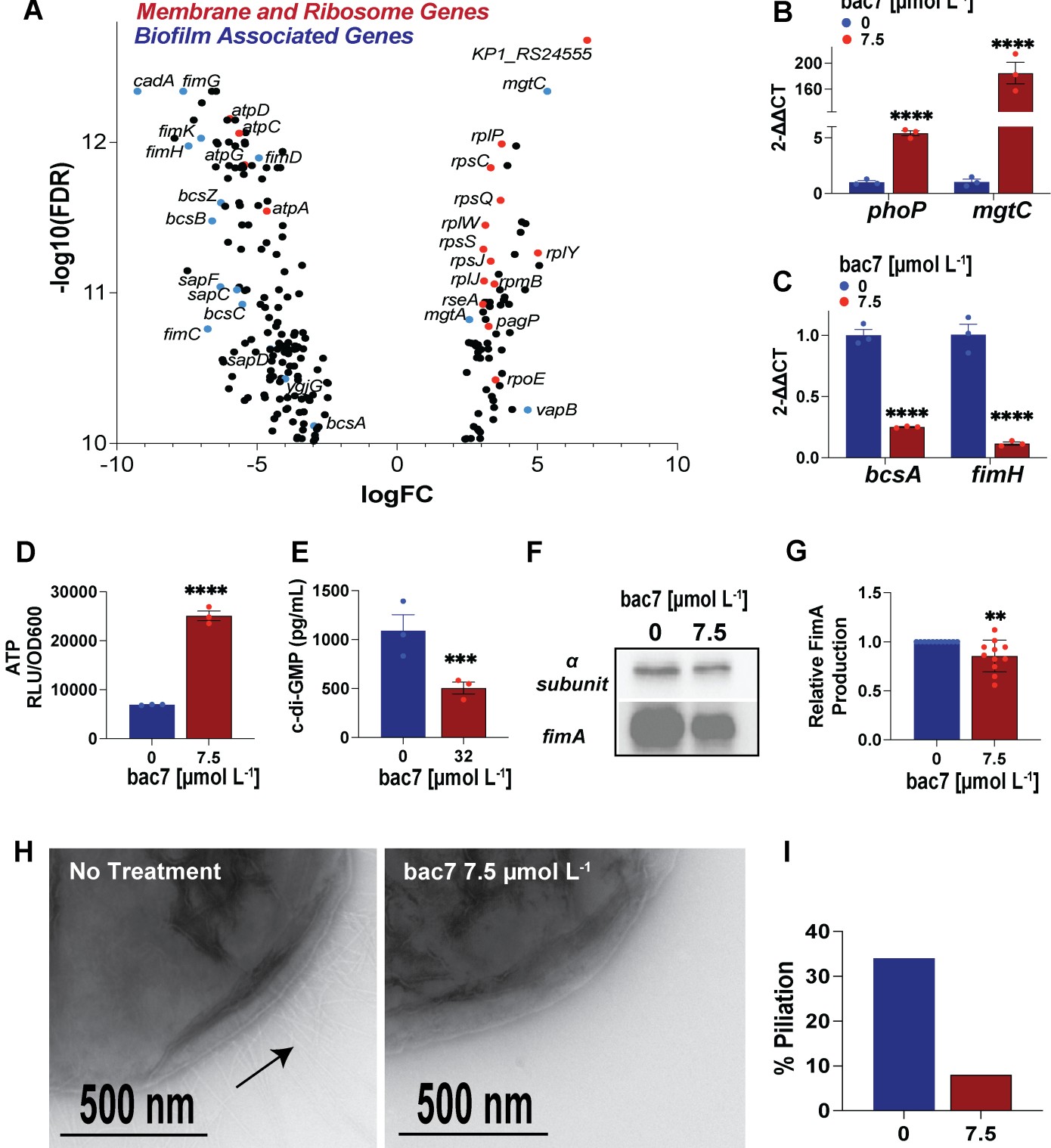

**Fig 2. Transcriptional insight reveals complex mechanism of killing and biofilm disruption.** Volcano plot of hvKp NTUH-K2044 most significantly differential expressed genes (DEGs) ($-\log_{10}$ FDR > 10) following 7.5 µmol L$^{-1}$ bac7 for 30 minutes **(A)**. Red dots show gene changes relevant to the

mechanism of killing and blue dots show biofilm important gene changes. RT-qPCR validation of *phoP* and *mgtC* (B), and c-di-GMP regulated *bcsA* and *fimH* (C) with 7.5 µmol L$^{-1}$ bac7 for 30 minutes. Intracellular ATP assessment following 7.5 µmol L$^{-1}$ bac7 for 30 minutes measuring relative luminescence units (RLU) using a CellTiter-Glo kit and normalized to the respective OD$_{600}$ (RLU/OD$_{600}$) (D). Intracellular c-di-GMP levels following 7.5 µmol L$^{-1}$ bac7 for 30 minutes using a Cyclic di-GMP ELISA kit (E). The RT-qPCR was normalized using a *ftsZ* housekeeping gene. One-way ANOVA was used to determine significance compared to the no treatment groups for B-E with Dunnett's correction for multiple comparisons with adjusted p-values shown (asterisks indicate p-values **** < 0.0001, ** < 0.01, and ns > 0.1) and error shown reported as ±SEM. Immunoblot of hvKp NTUH-K2044 with 7.5 µmol L$^{-1}$ bac7 treatment using cultures normalized by optical density showing FimA antibody staining next to RNA polymerase alpha subunit control (F). The relative abundance of FimA from n = 11 western blots was quantified using ImageJ with error shown as ±SD and significance determined by unpaired t-test (p-value 0.0076) (G). Transmission electron microscopy (TEM) of hvKp NTUH-K2044 with or without 7.5 µmol L$^{-1}$ bac7 using antibody labeling of FimA fimbriae (H). Percent piliation was determined by manually counting at least 50 cells/sample and dividing the number of cells with extruding pili by the total number of cells counted in the TEM images for I.

## Bac7 transient membrane localization induces hvKp membrane depolarization but not leakage

Our RNA sequencing revealed bac7 increased the expression of genes encoding the membrane stress response protein RpoE and ribosomal subunits, together suggesting membrane and ribosomal stress are induced. Although bac7 has traditionally been described as a non-lytic peptide that kills via protein synthesis inhibition, Runti et al. described that this peptide has various modes of action depending on the bacterial genera [36]. We hypothesize that although bac7 kills primarily via ribosomal inhibition, membrane stress is a contributing factor to the mechanism of killing and biofilm eradication in *K. pneumoniae*.

To determine if bac7 leaves the *K. pneumoniae* bacterial membrane intact similar to that observed with *S. enterica* and *E. coli* bacterial membranes [35], we tested bac7 in a β-galactosidase leakage assay using o-nitrophenyl-β-d-galactopyranoside (ONPG) and measured the cytosolic pH using pHrodo dye to assess for leakage of hydrogen ions. We found no leakage (S5A and S5B Fig) or changes in pH (S5C Fig) with bac7 treatment confirming its non-lytic nature. We then assessed membrane polarity with the hydrophobic cationic dye 3,3'-Dipropylthiadicarbocyanine Iodide (DiSC$_3$) in a dose-response analysis over the course of 30-minutes, following a 30-minute pre-incubation with DiSC$_3$ to allow for the dye to quench into the membrane. We found concentrations of 0.95, 1.9, and 3.8 µmol L$^{-1}$ of bac7 depolarized the inner membrane of hvKp NTUH-K2044 within the first 10 minutes after the addition of the peptide, shown by the sharp increase in fluorescence following the addition of bac7 (Fig 3A). This was then followed by a subsequent decrease in fluorescence intensity indicative of the re-quenching of the dye into the inner membrane. The fluorescence intensity induced by bac7 was not as high or as sustained as the positive control membrane lytic peptide cecropin A but well above the negative control non-lytic small molecule ertapenem (S6 Fig). We saw similar bac7-mediated membrane depolarization with another hvKp strain KPPR1S (K2 capsule serotype) (Fig 3B). To take into account the role of capsular polysaccharides in the depolarization potential of bac7, we tested the capsular mutant KPPR1S Δ*wcaJ* [63,64] that has an increased MIC (0.8 µmol L$^{-1}$) compared to the parental isolate (0.24 µmol L$^{-1}$) and found slightly more membrane depolarization at 1.9 and 3.8 µmol L$^{-1}$. (Fig 3C). Similarly, when testing a colistin-resistant cKp strain MKP103, we found the membrane to have increased depolarization with bac7 treatment and minimal re-quenching (Fig 3D). Opposed to what we observed with hvKp NTUH-K2044, MKP103 showed slightly less depolarization with cecropin A lytic control than bac7 (S6 Fig). We next used high resolution single-cell fluorescence microscopy with FM-64 membrane dye and DAPI DNA stain to visualize the localization of a FITC N-terminal labeled bac7 peptide (FITC-bac7 MIC 1.9 µmol L$^{-1}$). We found that with 15 minutes of incubation with 7.5 µmol L$^{-1}$ of FITC-bac7, the peptide was enriched on the cellular envelope of hvKp NTUH-K2044 and KPPR1S as shown by the green localization with the red membrane dye, but not the KPPR1S Δ*wcaJ* capsular mutant (Figs 3E and S7A). Following 45 minutes incubation, we found, regardless of capsule abundance, the peptide had translocated into the cytosol as we see green fluorescence in the cytosol co-localized with the blue DAPI DNA stain rather than localized to the red membrane. However, when testing bac7 at the MIC concentration (1.9 µmol L$^{-1}$) we found at 15 minutes the peptide concentrated to the membrane of KPPR1S only (S7B Fig). We then performed TEM of hvKp NTUH-K2044 using a phosphotungstic negative stain to visualize the capsule (dark halo around cell) and membrane

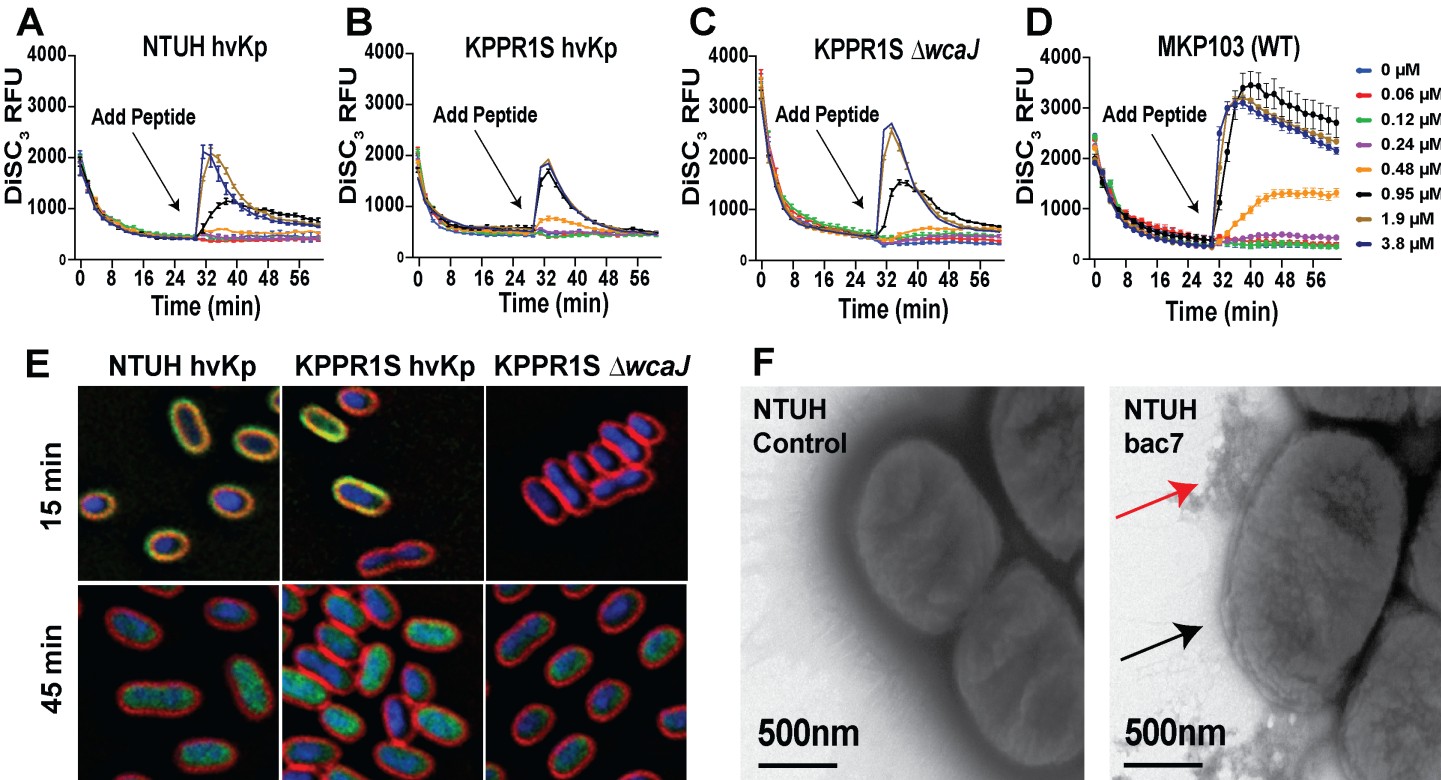

**Fig 3. Bac7 transient membrane localization induces hvKp membrane depolarization but not leakage.** Membrane depolarization of hvKp NTUH-K2044, hvKp KPPR1S, KPPR1S Δ*wcaJ*, and cKp MKP103 using a DiSC$_3$ membrane potential dye **(A-D)**. The dye was quenched into the membrane for 30 minutes before adding bac7 2-fold dilutions and the arrow indicates the peptide addition. Single cell fluorescent microscopy with FM 4-64 membrane dye (red), DAPI DNA stain (blue), after 15 minutes and 45 minutes incubation with 7.5 µmol L$^{-1}$ FITC-bac7 (green) **(E)**. Transmission microscopy of hvKp NTUH-K2044 with a 0.1% phosphotungstic negative stain with or without 7.5 µmol L$^{-1}$ bac7 **(F)**. Black arrow indicates membrane blebbing locations with treatment and red arrows show capsule deformation. Error for A-D are reported as ±SEM.

following treatment with 7.5 µmol L$^{-1}$ of bac7. This analysis revealed that although the membrane was largely intact, there was membrane blebbing (black arrows) indicating membrane stress ([Fig 3F]). Furthermore, although there was not a loss of capsule, we observed capsular distortion (red arrows) ([Fig 3F]). Our results show that, although bac7 does not cause cytosolic leakage and pore formation like traditional host-defense peptides [65], its passage through the membrane to access the cytosol causes depolarization of the bacterial membrane leading to a membrane stress response.

## Bac7 displays differential membrane depolarization across *K. pneumoniae* clinical isolates

We hypothesize transient depolarization occurs as the peptide interacts with the membrane while entering the cytosol as we observed in our fluorescence microscopy. Due to the variability between our hvKp and cKp lab strains ([Fig 3]), we hypothesize membrane depolarization variations between isolates suggests a differential rate of membrane translocation. To test this hypothesis, we wanted to expand our analysis beyond hvKp to broadly investigate membrane depolarization and FITC-bac7 uptake using clinical isolates from the *K. pneumoniae* MRSN diversity panel. For this, we performed DiSC$_3$ membrane depolarization assays with bac7 as described with the lab strains and selected isolates to perform high resolution single-cell fluorescence microscopy with FITC-bac7.

We assessed the depolarization profiles of the MRSN isolates that displayed high mucoviscosity (N = 14 lacking *rmpACD*; N = 3 carrying *rmpACD*) and strong biofilm formation potential (N = 15) in our previous work with these isolates

[15]. We found a diverse range of membrane depolarization of the MRSN isolates with bac7 and cecropin A, but as expected little depolarization with the beta lactam ertapenem (S8–S13 Figs). To facilitate a broad comparison of the MRSN isolates, we graphed DiSC$_3$ relative fluorescence at the lowest concentration that depolarizes the membrane (0.95 µmol L$^{-1}$ bac7) after 30 minutes treatment of bac7 (Fig 4A) or cecropin A and ertapenem controls (S14 Fig). Overall, we did not find a correlation between percent mucoviscosity that was determined in our previous study [15], and depolarization values that were determined in this study (SLR, p-value 0.6119, $R^2$ = 0.0087) (S15A Fig). However, only three isolates have the *rmpADC* operon, indicating sedimentation resistance in these isolates may be due to an independent factor. Interestingly, the two most mucoid isolates MRSN 21352 and 607210, have membranes that are very differently depolarized by bac7, although their mucoidy is unlikely due to hyper-capsule as they lack the *rmpADC* operon that contributes to mucoidy [7]. Therefore, we assessed whether colistin resistance determined in our previous work [66], correlated with increased depolarization and found a positive correlation between colistin MICs and membrane depolarization (SLR, p-value <0.001, $R^2$ = 0.1951) (S15B Fig). To understand how the differential membrane depolarization of *K. pneumoniae* affects the killing kinetics of bac7 we performed a time-kill assessment at the respective 0, 0.5, 1, and 4X MICs (Table 2) of select MRSN isolates (MRSN 365679 and 1912) that displayed different levels of membrane depolarization. As expected from a protein synthesis inhibitor, we found bac7 to have bacteriostatic activity rather than bactericidal activity, although we did observe a shift to bactericidal activity after 24-hours with higher concentrations of bac7 towards MRSN 365679 (S16 Fig).

We then used isolates displaying either high (MRSN 365679 and 21352) or low (MRSN 607210 and 1912) membrane depolarization to test localization of a FITC-bac7 peptide after 15 minutes at 1.9 µmol L$^{-1}$ using high resolution single-cell fluorescence microscopy as done with our hvKp isolates in Fig 3E. Although we did not see the same enrichment of FITC-bac7 to the cell envelope that we observed with our hvKp strains, we found a heterogenous localization pattern where a few cells displayed a high intracellular concentration of FITC-bac7, shown by an extreme green fluorescence inside the cells (Fig 4B). Interestingly, we saw an inverse correlation between membrane depolarization and peptide uptake where MRSN 365679 and 21352 displayed increased membrane depolarization but had fewer cells with concentrated peptide (n = 5 and n = 7, respectively), compared to MRSN 607210 and 1912 that had less membrane depolarization by bac7 but more cells with concentrated peptide (n = 26 and n = 10, respectively). Furthermore, cells with dense concentrations of peptide show cell rounding or have condensation of the cytosolic components. To understand this heterogeneity at a population level we performed flow cytometry analysis with MRSN 365679 (high depolarization/low uptake) and 607210 (low depolarization/high uptake) using *Bac*Light red cell stain following treatment with 1.9 µmol L$^{-1}$ FITC-bac7 for 30 minutes. In line with our fluorescence microscopy imaging results, we found compared to MRSN 607210, MRSN 365679 had less cells with an extreme abundance of FITC-bac7 uptake shown by increased proportion of cells within quadrant 2 (Q2) (Fig 4C and 4D). Overall, we found an inverse correlation between bac7 cytosol localization and depolarization of the bacterial membrane, indicating LPS membrane modification leading to colistin resistance in *K. pneumoniae* can change the rate of bac7 penetration to the cytosol.

## Spatial polysaccharide distribution and cellular density generate biofilm mediated resistance in clinical isolates

Our previous work revealed the potential of bac7 to collapse pre-formed biofilms of hvKp NTUH-K2044 [38]. To expand on this and understand the broad effects of this peptide towards isolates with extremely robust biofilm formation capabilities, we assessed the anti-biofilm capabilities of bac7 towards four MRSN clinical isolates that have strong biofilm formation potential [15]. We compared MICs to biofilm eradication of hvKp NTUH-K2044 and the MRSN isolates to measure the resistance attributed to biofilm formation. For this we used Calgary Biofilm Device (CBD) plates for biofilm formation with crystal violet staining to define 90% minimal biofilm eradication concentration (MBEC$_{90}$) of bac7 and clinical antibiotics. We found varying levels of biofilm mediated resistance (low MICs and high MBEC$_{90}$s) (Table 2) with all clinical isolates displaying an in increased fold change in biofilm mediated resistance compared to NTUH-K2044 (S17A Fig).

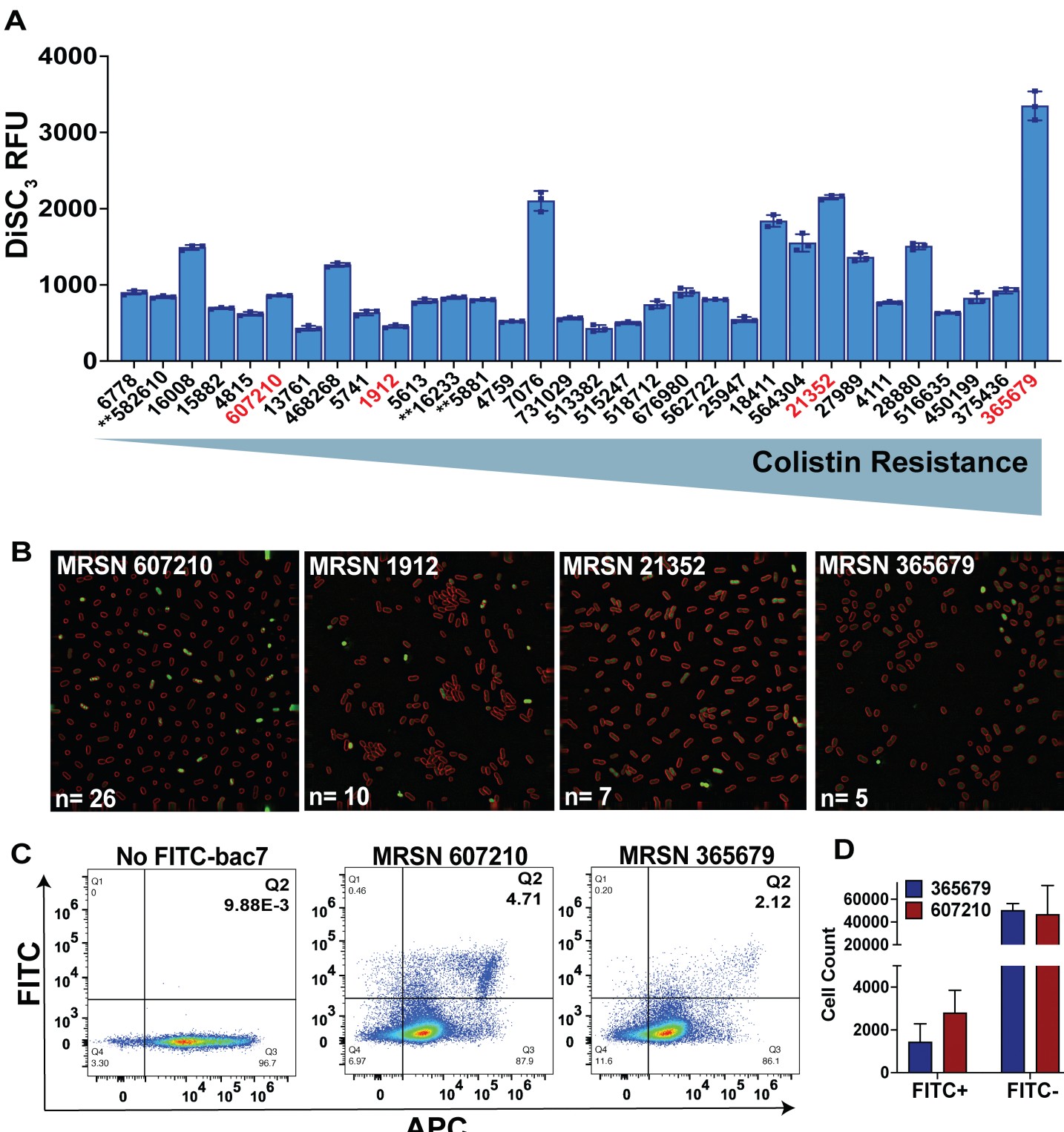

**Fig 4. Bac7 displays differential membrane depolarization across *K. pneumoniae* clinical isolates.** The figures show the membrane depolarization and FITC-bac7 uptake using the MRSN clinical diversity panel isolates. DiSC$_3$ fluorescent readings after addition of 0.95 µmol L$^{-1}$ bac7 for 30 minutes with the isolates ordered from low to high colistin resistance (left to right), respectively based on MICs determined in our previous work **(A)**. Mucoid

isolates that harbor the *rmpADC* locus are marked with two asterisks (**) and isolates with red text were chosen for the single cell fluorescent micros-copy. Single cell fluorescent microscopy with FM 4-64 membrane dye (red) after 15 minutes incubation with 1.9 µmol L$^{-1}$ FITC-bac7 (green) **(B)**. The number of cells (n) displaying high levels of FITC-bac7 uptake are shown for each isolate. Flow cytometry analysis with *Bac*Light red cell stain following 1.9 µmol L$^{-1}$ FITC-bac7 treatment of MRSN 365679 and 607210 for 30 minutes using FITC and APC lasers for green and red fluorescence, respec-tively **(C)**. Gating was performed to generate 4 quadrants (Q1: FITC+/APC-; Q2: FITC+/APC+ Q3: FITC-/APC+; Q4: FITC-/APC-) with a high bar set to quantify super FITC fluorescent cells in Q2. Quantification of the cells in quadrant 2 (FITC+/APC+) and quadrant 3 (FITC-/APC+) with triplicate flow cytometry samples **(D)**. Single flow cytometry figures shown as representative of triplicate experiments in **C**. Error for A and D are reported as ±SEM.

**Table 2. MIC and MBEC$_{90}$ values of clinical antibiotics and bac7 towards *K. pneumoniae* isolates.**

| | MIC/MBEC$_{90}$ [µmol L$^{-1}$] | CHL | ETP | GEN | PolyB | Bac7 |
|---|---|---|---|---|---|---|
| **NTUH-K2044** | MIC | 25 | ≤ 2 | ≤ 2 | 0.75 | 0.24 |
| | MBEC$_{90}$ | 50 | 4 | ≤ 2 | 3 | 3.8 |
| **MRSN 731029** | MIC | 25 | ≤ 2 | 134 | 0.75 | 0.48 |
| | MBEC$_{90}$ | ≥ 198 | 34 | ≥ 134 | 24.5 | ≥ 15 |
| **MRSN 564304** | MIC | 12 | ≤ 2 | ≤ 2 | 1.5 | ≤ 0.06 |
| | MBEC$_{90}$ | ≥ 198 | 67 | 4 | 49 | ≥ 15 |
| **MRSN 1912** | MIC | 25 | ≤ 2 | ≤ 2 | 0.75 | 0.24 |
| | MBEC$_{90}$ | ≥ 198 | 34 | ≤ 2 | 24.5 | 3.8 |
| **MRSN 16008** | MIC | 25 | ≤ 2 | ≤ 2 | 0.38 | 0.12 |
| | MBEC$_{90}$ | 50 | ≤ 2 | 134 | 12 | 7.5 |

*Abbreviations used for clinical antibiotics: chloramphenicol (CHL), ertapenem (ETP), gentamicin (GEN), polymyxin B (PolyB), and bac7 (1–35) (Bac7).

Our results show strong biofilm mediated resistance to bac7 treatment with MRSN 564304 (Table 2 and S17B Fig). To continue to investigate the resistance displayed by MRSN 564304 compared to the other isolates tested, we assessed the biofilm matrix spatial distribution of the four MRSN isolates using confocal z-stack imaging with SYTO 9 cellular stain (green) and calcofluor white polysaccharide stain (blue) to visualize the 3D spatial distribution of the cellular and polysac-charide populations, respectively [15]. This technique is unique from crystal violet staining due to the ability to facilitate measuring of biofilm height and differentiation of the cells from matrix material. We found that 15 µmol L$^{-1}$ of bac7 was able to collapse MRSN 1912 and 16008 (S17C–S17D Fig) more effectively than MRSN 731029 and 564304 (S17E and S17F Fig). We then tested the potential for bac7 adjuvant therapy with the protein synthesis inhibitor chloramphenicol that binds to the 50S ribosomal subunit and is impacted by biofilm mediated resistance (S17E and S17F Fig and Table 2) [67]. We found that although bac7 or chloramphenicol (198 µmol L$^{-1}$) alone were not effective against MRSN 564304 and 731029 biofilms, when combined they effectively collapse these extremely robust biofilms.

We hypothesized that penetration may be impacting bac7 biofilm disruption of MRSN 564304 biofilms. To test this hypothesis, we treated pre-formed biofilms with FITC-bac7 and used confocal z-stack imaging with BactoView red nucleic acid stain (red bacterial cells) and calcofluor white dye (blue polysaccharide matrix), to visualize the localization of the FITC-bac7 peptide (green peptide). With one-hour treatment FITC-bac7 could not only penetrate the biofilm of a sensitive isolate MRSN 1912 (Fig 5A), but also the resistant biofilm of MRSN 564304 (Fig 5B). To understand the role of cell density of the resistant biofilm compared to the sensitive biofilm we quantified the cellular population and peptide mean fluores-cence intensity using the BiofilmQ image processing tool [68]. This software tool is used to segment the confocal z-stack images to allow for spatial fluorescence quantification. We found the mean fluorescence of FITC-bac7 fluorescence more closely matched the red cell mean fluorescence with MRSN 1912 (Fig 5C) than with MRSN 564304 (Fig 5D) indicating the relative peptide concentration does not cover the cell density of this robust biofilm. To understand if the cells within the

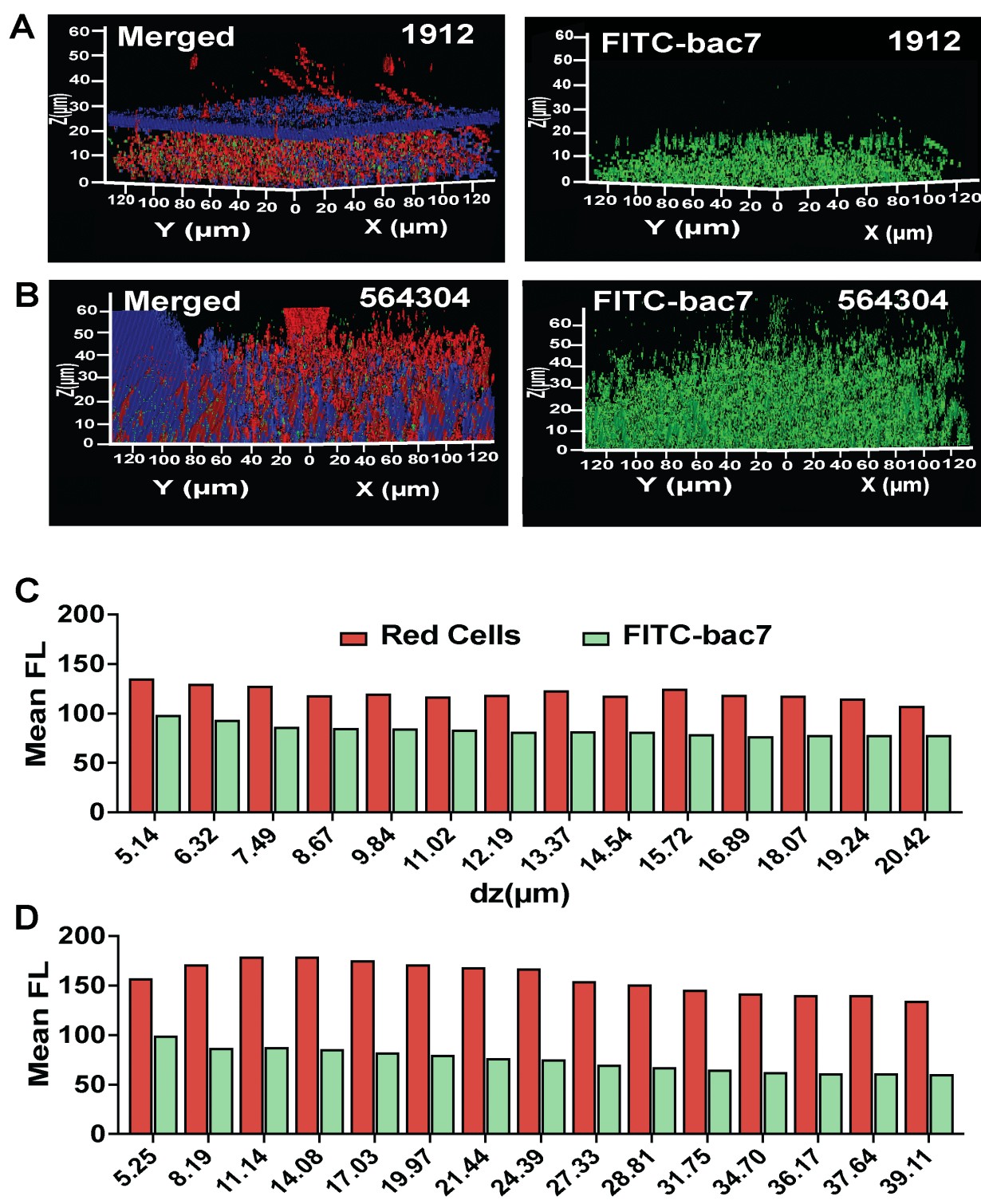

**Fig 5. Spatial polysaccharide distribution and cellular density generate biofilm mediated resistance in clinical isolates.** The figures show confocal z-stack images of FITC-bac7 treated MRSN 1912 and 564304 biofilms. A and B shows 1 hour treatment of MRSN 1912 and 564304, respectively with 15 µmol L$^{-1}$ FITC-bac7, stained with BactoView cell stain (red) and calcofluor white polysaccharide stain (blue) to visualize localization of the FITC-bac7 peptide (green). All imaging was performed in triplicate with one representative image shown. BiofilmQ software was used to process the confocal z-stack images and graph the mean relative fluorescence in each z-stack layer (dz (µm)) for red labeled cells and green FITC-bac7 peptide **(C and D)**.

biofilm of MRSN 564304 were being affected by bac7 we measured the levels of ATP in biofilm growing cells as a metric of peptide reaching its cytosolic target. We saw a significant increase in ATP three hours post-treatment (S18A Fig) that aligned with a significant decrease in biofilm density compared to the no treatment group (S18B Fig). These results suggest that, although bac7 can penetrate the biofilm matrix of MRSN 564304 within one-hour, increased abundance of cells before treatment is likely the cause of the biofilm mediated resistance of this clinical isolate.

**Mutant analysis reveals both SbmA and MgtC are important for mitigating the biofilm disruption potential of bac7**

We next aimed to understand more about bac7 membrane interactions and the consequences of triggering *mgtC* using bacterial mutants lacking the SbmA transporter and MgtC, respectively. SbmA is the main transporter used by polyproline peptides to enter the bacterial cytosol and bac7 has been shown to bypass this transporter in a concentration dependent manner in other species [36]. We also show here with *K. pneumoniae,* at low concentrations the peptide relies heavily on the transporter, yet at elevated concentrations it causes membrane stress. Therefore, we hypothesized that a mutant lacking SbmA would show increased depolarization as the peptide will be only accessing the cytosol by membrane penetration. Furthermore, we hypothesized from *S. enterica* literature [45,61], that without MgtC there would be elevated ATP levels if this protein has a similar function in *K. pneumoniae*. To test these hypotheses, we used transposon mutants of cKp MKP103 parental strain, which will also test our hypothesis that the high membrane depolarization by bac7 (Fig 3D) is due to the slow penetration of this peptide into the cytosol of this colistin-resistant strain [69]. As expected with this colistin-resistant strain, we do see increased baseline expression of *phoP* and *mgtC* in normal LB growth conditions (without colistin pressure) compared to hvKp NTUH-K2044 and KPPR1S (Fig 6A).

When assessing ATP levels in cKp MKP103, interestingly we do not observe the same increased intracellular ATP after 30 minutes with 1.9 µmol L$^{-1}$ bac7 that we observed with hvKp NTUH-K2044 but found with this colistin-resistant strain that the RLU/OD$_{600}$ closely matched the no treatment group (1.36-Fold Change) (Fig 6B). This is in line with our hypothesis that the increased depolarization displayed by this colistin-resistant strain (Fig 3D) is resulting from the decreased rate of penetration into the cytosol. However, when comparing the Δ*sbmA* (KPNIH1_05310–803::T30) (MIC 3.8 µmol L$^{-1}$) mutant ATP levels to the parental strain (MIC 0.48 µmol L$^{-1}$), we observed decreased intracellular ATP, indicating the peptide is not reaching the ribosomal cytosolic target as well when the transporter is lost with this strain. When testing the Δ*mgtC* (KPNIH1_13815–409::T30) (MIC 0.48 µmol L$^{-1}$) mutant, we found increased intracellular ATP compared to the parental strain, indicating that MgtC functions in *K. pneumoniae* as shown previously in *S. enterica,* decreasing toxic levels of ATP [39,44,70]. Intriguingly, we found even with no treatment there was increased intracellular ATP with the Δ*mgtC* transposon mutant compared to the parental isolate (S19A Fig). When assessing levels of c-di-GMP with the parental cKp MKP103 and the Δ*mgtC* transposon mutant following treatment with bac7, there was a decrease in c-di-GMP compared to the no treatment with the parental isolate, but increased c-di-GMP with the Δ*mgtC* transposon mutant (Fig 6C). We observed similar results when testing the ribosomal binding control tetracycline (S18B and S19C Figs). We then tested the membrane depolarization abilities of bac7 against the Δ*sbmA* and Δ*mgtC* transposon mutants (S19D–S19G Fig). We found a concentration dependent change in the peak fluorescence intensity (10 minute time point) with Δ*sbmA,* where with 0.95 µmol L$^{-1}$ bac7 there was less depolarization of the membrane but more depolarization with 1.90 µmol L$^{-1}$ bac7 compared to the parental strain (Fig 6D) that correlates with the loss of ATP observed with this mutant with bac7 treatment at this concentration if sustained membrane interactions causes membrane leakage. When comparing Δ*mgtC* transposon mutant to the parental strain there was an overall increase in membrane depolarization, where the lowest concentration to cause depolarization was 0.12 µmol L$^{-1}$ (S19D Fig) compared to 0.48 µmol L$^{-1}$ with the parental strain (Fig 3D), indicating the membrane of this isolate is more sensitive overall to bac7 and suggesting a role of MgtC in maintaining the membrane integrity of *K. pneumoniae*.

We then tested bac7 biofilm disruption at 15 µmol L$^{-1}$ using crystal violet staining of MKP103 parental strain next to Δ*sbmA* and Δ*mgtC* transposon mutants and found both mutants were more sensitive to bac7 compared to the parental

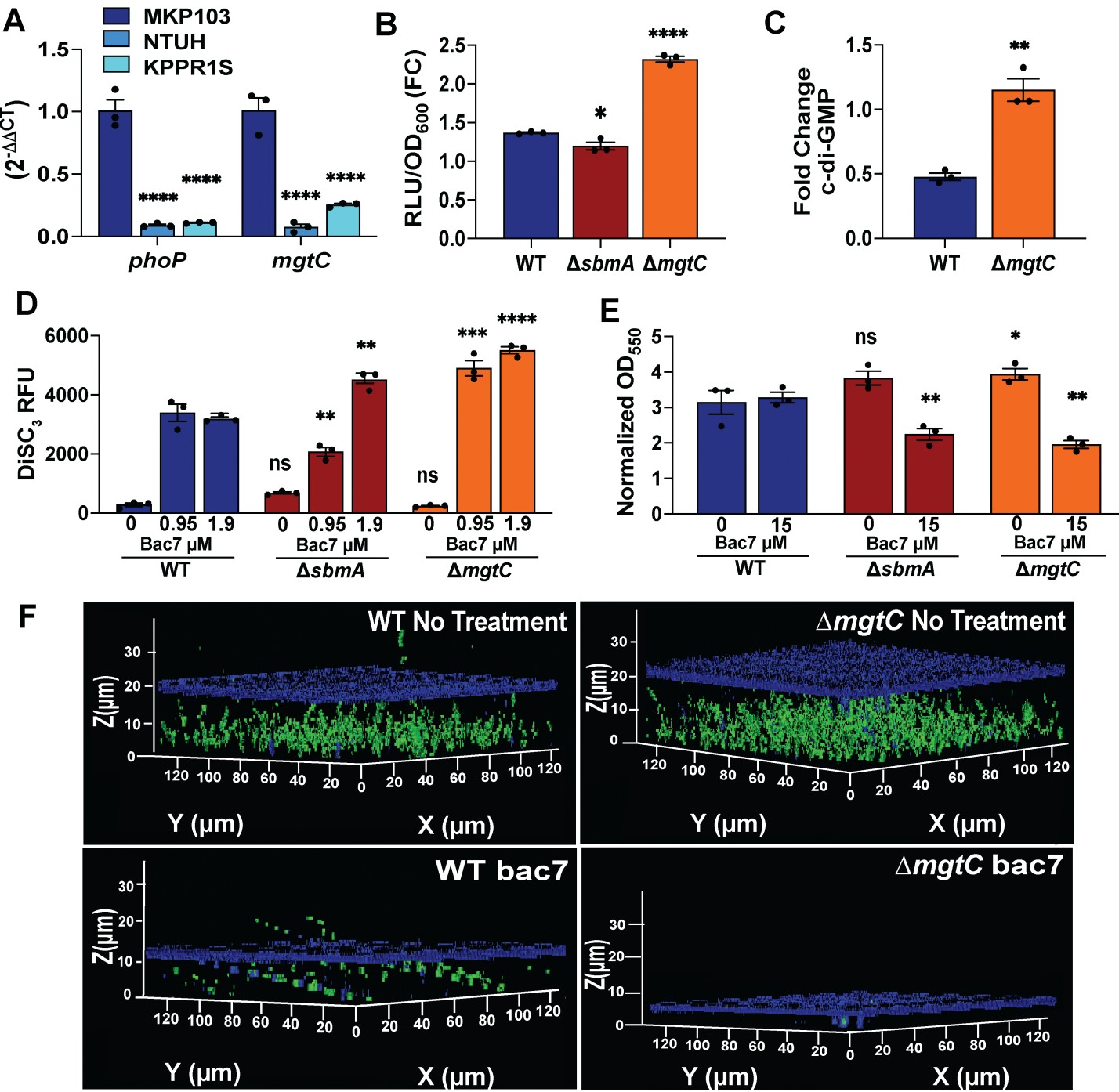

**Fig 6. Mutant analysis reveals both SbmA and MgtC are important for mitigating the biofilm disruption potential of bac7.** The figures show comparisons of the parental MKP103 (WT) and mutants lacking the SbmA polyproline transporter (ΔsbmA) and MgtC *(ΔmgtC)*. Relative gene expression of *phoP* and *mgtC* grown in LB to log phase to analyze baseline expression of these genes in colistin-resistant MKP103 parental isolate compared to hvKp NTUH-K2044 and hvKp KPPR1S using a 16S housekeeping gene for normalization **(A)**. Intracellular ATP levels shown as fold change in RLU/$OD_{600}$ between no treatment and 1.9 µmol $L^{-1}$ bac7 treatment for 30 minutes for parental MKP103 (WT), MKP103 ΔsbmA, and MKP103 ΔmgtC **(B)**. Fold change intracellular c-di-GMP levels between no treatment and 7.5 µmol $L^{-1}$ bac7 for MKP103 parental (WT) and MKP103 ΔmgtC quantified using the Cyclic di-GMP ELISA kit **(C)**. $DiSC_3$ membrane potential dye assessment of inner membrane depolarization with 0, 0.95, and 1.9 µmol $L^{-1}$ bac7 showing the peak fluorescence observed following 10 minutes treatment for parental MKP103 (WT), MKP103 ΔsbmA, and MKP103 ΔmgtC **(D)**. Biofilm density measurement using crystal violet staining showing normalized (subtract background) biofilm density ($OD_{550}$) after 24 hours with no treatment or 15

μmol L$^{-1}$ bac7 **(E)**. 3D-rendering of confocal z-stack images of pre-formed biofilms of the MKP103 parental isolate (WT) and Δ*mgtC* mutant untreated and treated with 15 μmol L$^{-1}$ bac7 **(F)**. Cells were stained with SYTO9 (green cells), and matrix polysaccharides were stained with calcofluor white (blue matrix). n = 3 biofilms imaged for each condition with a representative image shown. Two-way ANOVA was used to determine significance for A in comparison to MKP103 with Tukey's multiple comparison correction, and one-way ANOVA was used to determine significance for C-F comparing the mutants to the WT with Tukey's multiple comparison correction. Significance is shown with asterisks (adjusted p-values **** < 0.0001, *** < 0.001, ** < 0.01, * < 0.1, and ns > 0.1) and error shown as ±SEM.

strain (Fig 6E). To further investigate the Δ*mgtC* transposon mutant biofilms, we used confocal z-stack imaging with SYTO 9 (green) and calcofluor white (blue) dyes to visualize the biofilm changes in the mutant with and without peptide treatment. We found that without treatment Δ*mgtC* had increased biofilm cell density compared to the parental strain (Fig 6F). However, in line with the crystal violet staining, there is increased disruption of the Δ*mgtC* biofilm compared to the parental strain MKP103. Collectively, these results suggest bac7 is dependent on the *K. pneumoniae* SbmA transporter at low concentrations and that *K. pneumoniae* MgtC is functioning to relieve toxic ATP levels.

### *In vivo* treatment decreases bacterial burden and colonization of vital organs in a skin abscess biofilm infection model

To assess the ability of bac7 to treat a clinically relevant *K. pneumoniae* biofilm, we leveraged a murine model of hvKp skin abscess. To enable biofilm formation prior to treatment, we waited five hours after bacterial inoculation before treatment (peptide or the commonly used peptide positive control polymyxin B) administration. Mice with a pre-formed linear skin abrasion were infected with $1 \times 10^4$ colony forming units (CFU) of hvKp NTUH-K2044 (20μl of a $5 \times 10^5$ CFU mL$^{-1}$ solution). Five hours post-infection, a single dose of bac7 (20 μmol L$^{-1}$; 10 × MIC) or polymyxin B (10 μmol L$^{-1}$; 10 × MIC) was administered to the infection site. Mice were monitored for four days, with skin tissue and organs (liver, spleen, and kidneys) harvested on days two and four to quantify bacterial burden (Fig 7A). Mouse weight was recorded as a surrogate for disease severity (Fig 7B) [71]. Notably, only bac7 treatment resulted in weight gain over the course of infection. The polymyxin B treated mice lost weight compared to the no treatment control group, indicating the mice could be experiencing adverse side effects that have been documented with this antibiotic [72].

At day two post-treatment, bacterial burden in skin tissues was reduced in both treatment groups compared to the untreated control (S20 Fig), although colonization of the distal organs was minimal at this early stage ($10^2$-$10^3$ CFU mL$^{-1}$ compared to $10^6$ CFU mL$^{-1}$ in skin tissue samples). In contrast, four days post-treatment, colonization of the distal organs was more evident. Treatment with either bac7 or polymyxin B led to a reduction in CFU counts in the skin and distal organs (Fig 7C). Taken together, these results indicate that bac7 reduces local biofilm-associated bacterial burden, which limits colonization of hvKp in the distant organs.

## Discussion

Given the important contribution of biofilm formation to clinical infections and the extreme drug resistance observed in *K. pneumoniae*, targeting biofilms and understanding the underlying mechanisms of biofilm disruption are imperative steps towards better therapeutics. We previously described the biofilm disruption of hvKp NTUH-K2044 by bac7 and decreased mucoviscosity within the dispersed population [52]. We found strong peptide aggregation due to a large portion of the N-termini bound to polysaccharides resulting in peptide-polysaccharide aggregation and biofilm collapse [38]. Previous literature shows this peptide has different mechanisms of action between bacterial genera [36], but this had yet to be defined in *K. pneumoniae*. Here we characterize the mechanism of *K. pneumoniae* killing and biofilm disruption by bac7 and demonstrate a spectrum of biofilm disruption potential against a set of diverse clinical isolates. Our transcriptional data revealed when *K. pneumoniae* is exposed to the concentrations of bac7 necessary to disrupt biofilms oxidative phosphorylation was decreased in expression, ribosomal subunits are increased in expression, and there were changes in genes important for biofilm formation. Of note, we also found signs that at this elevated concentration, there is membrane

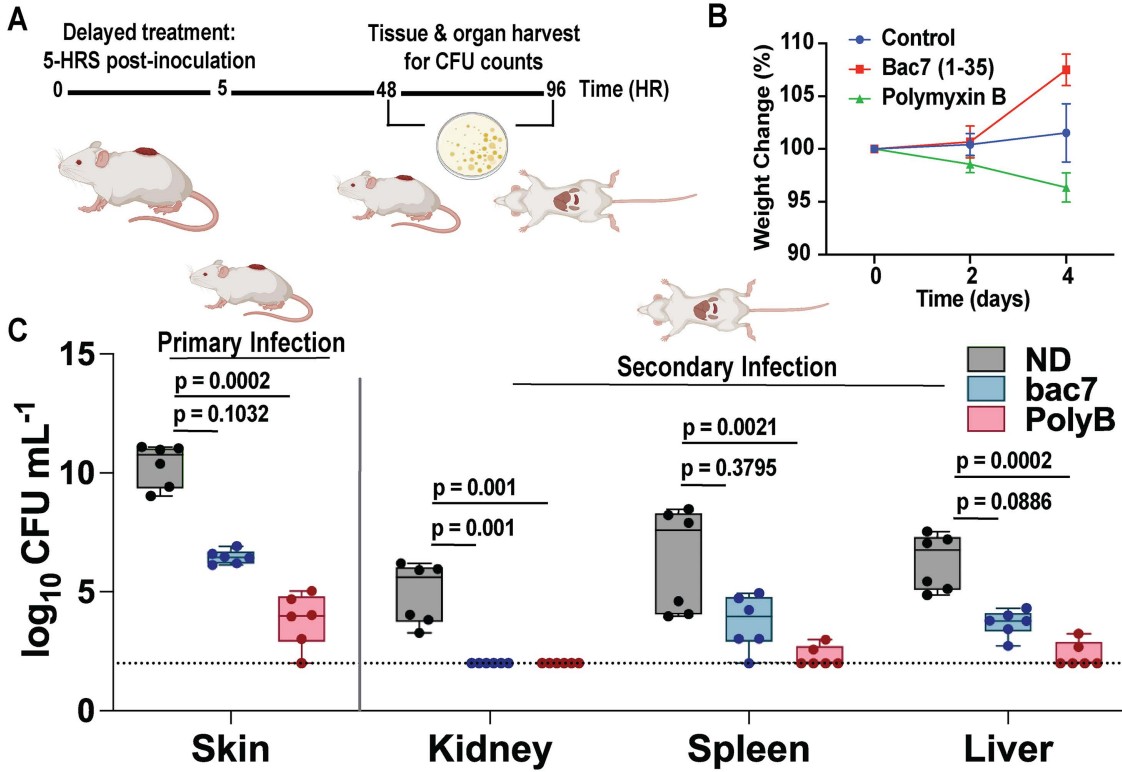

**Fig 7. *In vivo* treatment decreases bacterial burden and colonization of vital organs in a skin abscess biofilm infection model.** Skin abrasions on the ventral side of mice were formed and infected with 1x10⁴ CFU hvKp NTUH-K2044. 5 hours post-infection, a single dose of bac7 and positive control polymyxin B (10X MICs) was administered onto the infection site **(A)**. Four days post-treatment, regions of skin abrasions were excised, and organ systems were harvested and homogenized to quantify the bacteria. Mouse weight assessed at day two and day four post infection and shown as a percentage weight change compared to day 0 **(B)**. Log₁₀ CFU mL⁻¹ of the bacterial load in the skin, kidney, spleen, and liver **(C)**. Data was graphed as box plots to show the median values with error shown as ±SEM. The limit of detection (100 CFU mL⁻¹) is shown as a dotted line on the y-axis. Significance was determined using non-parametric Kruskal-Wallis one-way ANOVA of n = 6 mice with p-values were obtained by comparing untreated groups to the treated groups. BioRender was used to generate the vector images with the figure designed in Adobe Illustrator. Links for the BioRender vectors can be found: *Mouse with organs: Created in BioRender. Fleeman, R. (2025)* https://BioRender.com/i5yuud0. *Mouse vector: Created in BioRender. Fleeman, R. (2025)* https://BioRender.com/ackcbgr. *Created in BioRender. Fleeman, R. (2025)* https://BioRender.com/atzgcky. *Petri Dish: Created in BioRender. Fleeman, R. (2025)* https://BioRender.com/atzgcky.

stress with this non-lytic peptide indicating it has a dual killing mechanism, targeting both the membrane and ribosome. Intriguingly, we saw a gene encoding for an MgtC family protein was the second most increased in expression, and to date there are no studies into the function of this protein in *K. pneumoniae*.

The PhoPQ two component system facilitates a cascade of changes increasing outer membrane resistance to the last-resort antibiotic polymyxins [73]. Within the PhoP regulon is *mgtC*, found to be important for controlling toxic levels of ATP experienced in the low pH and low magnesium intramacrophage environment [45,61]. However, MgtC inhibition of Pi uptake limits this necessary component needed to produce c-di-GMP, an allosteric activator important for production of many important biofilm components and decreased c-di-GMP has been shown to be the molecular switch from sessile to motile phase in bacteria [74]. Studies in *S. enterica* reveal MgtC not only decreases the production of cellulose but also decreases the bacterial membrane potential [30,70]. MgtC has been shown to have a role in biofilm formation and exopolysaccharide production in *P. aeruginosa* [50]. However, the function of this protein has yet to be determined in *K. pneumoniae*. We found that both *phoP* and *mgtC* had increased expression in our transcriptional data although there were

extremely high levels of *mgtC* with bac7 treatment. Within the *mgtC* leader mRNA, hairpin autoregulation was shown to be induced by ATP [44]. Therefore, we predict that once transcript of *mgtC* is produced through activation of the PhoP regulon, the elevated ATP levels caused by bac7 binding to the ribosomes is inducing the extreme increase in *mgtC* transcription. This resulted in decreased expression of c-di-GMP allosterically activated *bcs* and *fim* operons and decreased FimA production. We hypothesize that this is the cause of the rapid switch from biofilm formation before a significant change in viability.

Bac7 is in a class of proline rich peptides that enter the cell via the inner membrane protein transporter SbmA [35]. However, some reports have shown membrane mechanism of action with this peptide towards *P. aeruginosa* compared to *E. coli* [51]. Our transcriptional analysis revealed there may be more than one mechanism towards *K. pneumoniae*. Our single cell fluorescence imaging to visualize the entry of bac7 into the cytosol revealed transient membrane localization that we show leads to membrane depolarization and damage (Fig 3). Furthermore, we found diverse membrane depolarization abilities between different clinical isolates, with a potential correlation with decreased cell penetration and membrane depolarization (Fig 4). In the colistin-resistant cKp MKP103 this increased membrane depolarization correlated with less ATP increase following bac7 treatment, revealing increased depolarization aligns with decrease in peptide entry. The MRSN diversity panel is a collection of clinical isolates recovered from the Walter Reed Army Institute of Research (WRAIR) from around the globe [75]. When assessing the diversity of MRSN clinical isolate membrane depolarization caused by our bac7, we found isolates with increased colistin resistance had overall higher membrane depolarization. Furthermore, we saw a heterogeneity with FITC-bac7 peptide uptake, where some cells had an abundance of peptide while others had very minimal. This combined with the increase in expression of the genes encoding the VapBC toxin-antitoxin system suggest the presence of a subpopulation of persister cells resisting peptide uptake. Future work will be targeted at understanding the potential for resistance development towards polyproline peptides through persistence or sleeper cells.

Although many host antimicrobial peptides have been shown to be transported through the SbmA transporter, the exact natural function of this transporter has yet to be established for enteric bacteria [76]. An ortholog of SbmA in *Brucella abortus*, BacA, has been shown to be important for intracellular infections due to its localization within an operon with *yaiW* [77]. YaiW is a palmitate-modified outer membrane lipoprotein shown to be involved in very-long-chain fatty acid modification in *Sinorhizobium meliloti* and *B. abortus* [78]. Although BacA has been shown to be important for lipid alterations in *B. abortus* and *S. meliloti*, SbmA was shown not to be involved in modifications to enteric LPS [78]. However, this work did not test *K. pneumoniae* and SbmA has been identified as a virulence factor in avian pathogenic *E. coli* and as a fitness factor in *K. pneumoniae* bloodstream infections [79,80], indicating more work needs to be done to define the role of this protein enteric pathogenesis. Overall, it is intriguing to consider that host defense peptides capitalize on a protein that is important for host virulence and that loss of this protein through spontaneous mutations to resist the peptide translocation would result in less virulent bacteria. Although not within the scope of this work, our future work aims to understand how spontaneous mutations in SbmA generated by serial passaging with bac7 effect the virulence of *K. pneumoniae*.

Our previous work focused on the K1 capsule serotype hvKp NTUH-K2044, while here we expanded our analysis to test the potential biofilm disruption ability of bac7 toward the MRSN panel of clinical isolates. We found bac7 was able to disrupt biofilms of MRSN 1912 and 16008 to a greater extent than MRSN 731029 and 564304. When testing biofilm mediated resistance of the clinical isolates, we found MRSN 564304 had strong biofilm-mediated resistance to bac7. Testing FITC-bac7 peptide penetration into biofilms, our confocal microscopy revealed penetration of the peptide into MRSN 564304 biofilm within one hour. However, using biofilmQ software we show the decreased peptide to cell ratio indicating insufficient cellular coverage. This shows the densely packed cells within MRSN 564304 biofilms requires more peptide to disrupt compared to MRSN 1912 that has less cell density within the biofilm matrix. The difference in cell density could also explain the slight difference in bac7 biofilm disruption between the strongest biofilm formers MRSN 731029 and 564304. Although both isolate biofilms are not completely disrupted by bac7, 731029 biofilms have overall less cell density without treatment compared to MRSN 564304 and consequently more sensitivity to bac7 treatment.

To provide a mechanistic insight into peptide penetration independent of SbmA and the role of MgtC in biofilm disruption we probed bacterial mutants lacking the SbmA transporter for peptide uptake and the MgtC like protein. We found bac7 does not induce the same levels of ATP in a colistin-resistant strain, but has elevated ATP in an Δ*mgtC* mutant, indicating MgtC is functioning as described in *S. enterica* [45,61]. When looking at the membrane depolarization potential we see more depolarization of the Δ*mgtC* mutant membrane at low concentrations but a concentration dependent increased depolarization with Δ*sbmA* where only at 1.90 μmol L$^{-1}$ do we see increased depolarization compared to the parental strain. This shows bac7 is more dependent on the transporter at lower concentrations validating the increased MIC of this mutant (1.9 μmol L$^{-1}$) while suggesting it is bypassing the membrane in a concentration dependent manner. Our future work will focus on this dosage dependent effect and how this effects the transcriptional changes observed in this study to determine more intricate stress pathways induced at lower concentrations. Moreover, the increased depolarization of the Δ*mgtC* mutant shows increased membrane sensitivity in this mutant compared to the parental strain, suggesting that in *K. pneumoniae* MgtC may also function to control the membrane potential as has been shown in *S. enterica* [70]. Future work will be focused on understanding the source and defining the consequences of the membrane depolarization induced by polyproline peptides. Finally, both mutants form a slightly stronger biofilm without peptide treatment compared to the parental strain but are more sensitive to bac7 treatment at 15 μmol L$^{-1}$. This shows that peptide interactions with the membrane are important for biofilm disruption and suggests that MgtC may play a role in protecting *K. pneumoniae* from bac7 biofilm disruption.

Finally, hvKp has emerged as a novel concern due to host immune evasion and dissemination of the bacterial infection to various organ systems through the bloodstream resulting in pyogenic liver abscess, endophthalmitis, renal abscess, and meningitis [6]. With the changes in gene expression, we modeled a *K. pneumoniae* biofilm abscess infection. With the loss of matrix material and decreased mucoidy of the dispersed population observed in our previous work, we hypothesized that this would decrease colonization of the kidneys, liver, and spleen by hvKp. Although bacterial barcoding is necessary to study dissemination dynamics [81], we demonstrate the ability of bac7 to decrease the bacterial burden in the wound and to decrease colonization of the kidneys, liver, and spleen (Fig 7). Intriguingly, the bac7 treated mice, as opposed to the polymyxin B mice, gained weight throughout the four-day experiment indicating the increased health with this treatment comparatively. This is an important factor considering the clinical toxicity shown by polymyxins [72]. Although preliminary, these findings suggest the potential of bac7 to prevent survival in primary and secondary sites using a wound infection model. Future studies will focus on using bacterial barcoding to determine how peptide treatment influences *K. pneumoniae* dissemination dynamics.

In conclusion, we have shown the dual action of bac7 can disrupt biofilms formed by hvKp and MDR cKp pathotypes, including clinical isolates. We reveal bac7 treatment potentially switches *K. pneumoniae* from biofilm state through c-di-GMP mediated regulation of cellulose and fimbriae. While this would be less advantageous for a non-antimicrobial, bac7 also imparts strong antibacterial activity towards the cells released from the biofilm for a unique combination of anti-biofilm and anti-bacterial mechanisms. Using a delayed treatment *in vivo* skin abscess model that emulates a biofilm mediated infection, we demonstrate that bac7 is a promising therapeutic for decreasing colonization of the kidneys, liver, and spleen of biofilm-associated hvKp wound infections.

## Methods

### Ethics statement

The skin abscess infection mouse model was revised and approved by the University Laboratory Animal Resources (ULAR) from the University of Pennsylvania (Protocol 806763).

### Bacterial strains

Bacterial lab strains used in this study are listed in S2 Table. The Walter Reed Army Institute of Research Multidrug-Resistant Organism Repository and Surveillance Network (MRSN) diversity panel of *K. pneumoniae* isolates were used

for the clinical isolates in this study are listed in (S3 Table). All overnight cultures were grown in Luria-Bertani (LB) broth at 37°C with 220 rpm shaking. Overnight cultures were synchronized to log phase using 1:10 subculturing into LB broth and grown for 3 hours 37°C with 220 rpm shaking. Chloramphenicol 100 μg mL$^{-1}$ was supplemented to LB when growing cKp MKP103 transposon mutants.

## Peptides

Bac7 (1–35) peptide was ordered from Novopro (novoprolabs.com/p/bac7) and polymyxin B sulfate from TCI chemicals (https://www.tcichemicals.com). Novopro synthesized the N-terminal FITC tagged bac7 peptide used for the microscopy and flow cytometry analyses. All peptides were resuspended in ultra-purified water at 10 mg mL$^{-1}$ and stored at -20 °C in 50 μL aliquots.

## Transcriptional analyses

**Bac7 treatment and RNA extraction.** Overnight cultures of hvKp NTUH-K2044 were synchronized for 3 hours diluted to an optical density 600 nm (OD$_{600}$) 0.5 in Mueller Hinton Broth 1 (MHB1), treated with 7.5 μmol L$^{-1}$ bac7 for 30 minutes in a water bath at 37 °C, in parallel with no treatment controls. After incubation, the samples where centrifuged at 15,000 × g and processed for total RNA extraction using the hot phenol procedure as previously described with modifications [22]. Briefly, RNA was resuspended in 10 μl nuclease free water and quantified by Nanodrop. The RNA was then DNase treated using TURBO DNA-*free* Kit (AM1907, Invitrogen). RNA samples were aliquoted and stored at -80 until downstream analysis by RNA sequencing and RT-qPCR.

**RNA-sequencing.** The RNA samples were submitted to SeqCenter (https://www.seqcenter.com/) and sequenced using a NovaSeq X Plus System. Read quantification was performed using Subread's feature Counts functionality [82]. After normalization, read counts were converted to counts per million (CPM). Differential expression analysis was performed using edgeR's glmQLFTest. Differentially expressed genes (DEGs) were identified as the genes with log2FC > 1 and FDR < 0.01 between the bac7 treated samples and no treatment controls. Pathway analysis was performed using limma's "kegga" functionality [83]. The genes that were considered Up/Down in this analysis were at FDR < 0.05

**Protein–protein interaction (PPI) network construction.** A subgroup of annotated DEGs after a stringing filtering by Log2FC and FDR (Log2FC > 1.5, FDR < 10e$^{-10}$) for a total of 216 genes were uploaded to the Search Tools for the Retrieval of Interacting Genes (STRING, http://www.string-db.org/) [84]. In the PPI network analysis, a confidence > 0.9 was defined as the cut-off criterion.

**Reverse transcriptase quantitative PCR (RT-qPCR).** RNA samples were reverse transcribed using SuperScript IV VILO Master Mix (11756050, Invitrogen) following the manufacturer's instructions. RT-qPCR was performed using PowerUp SYBR Green Master Mix (A25742, Applied Biosystems) following the manufacturer's instructions, and the QuantStudio 3 Real-Time PCR System (A28567, Applied Biosystems). There were three technical replicates against three biological replicates for qPCR analysis. The primers used for RT-qPCR were designed using the Integrated DNA Technologies (IDT) PrimerQuest tool. Primer sets used are listed in S4 Table. Relative expression levels of the target transcripts were calculated using the 2$^{-\Delta\Delta Ct}$ method [85]. The genes *ftsZ, rpoD,* and 16S rRNA were used as endogenous housekeeping genes. Two-way ANOVA was used to determine significance of the gene expression changes.

## Intracellular ATP quantification

**Planktonic.** Overnight cultures were grown in Luria-Bertani (LB) media at 37°C in shaking incubator (220 rpm) for 24 hours. Overnight cultures were synchronized 1:100 in LB media for 3 hours in shaking incubator at 37°C. Synchronized cultures were standardized to OD 0.5 in MHB1 media and 2 mL culture was pipetted into culture tubes with 30 × MIC of the controls polymyxin B and chloramphenicol or 15 μmol L$^{-1}$ bac7. Cultures were incubated in shaking incubator (220 rpm)

at 37°C for 30 minutes. Cultures were centrifuged at 21,130 × g for 10 minutes and pellets were resuspended in 1 mL LB media. Samples were heat-killed in the 70°C water bath for 1 hour 100 µL of samples were pipetted into a 96-well black-walled clear bottom plate (Corning) with 100 µL CellTiter-Glo 2.0 Cell Assay reagent and placed into a BioTek SYNERGY H1 plate reader for 2 minutes of orbital shaking followed by 10 minute incubation. Luminescence was read after 10 minutes incubation. Results were measured in relative luminescence units (RLU) and normalized to the respective $OD_{600}$ values. Fold change values were achieved by dividing RLU/$OD_{600}$ of the treated samples by the RLU/$OD_{600}$ of the no-treatment control samples and graphed with error reported as ±SEM.

**Biofilm.** Biofilms were grown and treated as described for the confocal imaging above and incubated in 37°C static incubator for 30 minutes or 3 hours. Dispersed cells and biofilms were collected into 1.5 mL microcentrifuge tubes and heat-killed at 70°C for 1 hour in water bath. 100 µL heat killed biofilm and dispersed biofilm supernatant were analyzed as described above for the planktonic treatment cultures.

## c-di-GMP quantification

Quantification of c-di-GMP was performed using a Cyclic di-GMP ELISA (Cayman Chemical). Overnight cultures of hvKp NTUH-K2044, MKP103 WT, and MKP103 Δ$mgtC$ were synchronized for 3 hours and treated with 7.5 µmol L$^{-1}$ bac7 for 30 minutes at 37°C with 220 rpm shaking. The cultures were centrifuged at 15,000 × g and resuspended in Wash Buffer (#400062) with 0.5 µg mL$^{-1}$ polysorbate. Samples were then sonicated on ice for 20 seconds with 59 seconds delay for a total of 15 minutes and centrifuged at 15,000 × g for 5 minutes at 4°C. The assay was then performed according to kit instructions (Caymanchem.com/product/501780/cyclic-di-gmp-elisa-kit). Briefly, Cyclic di-GMP ELISA Standards (#401784) were made by diluting the bulk standard 1:3 into Immunoassay Buffer C (1×) (#401703). Tracer dye (#401780) was made by adding 60 µL of dye into 6 mL tracer for a 1:100 final dilution. Antiserum Dye (#401782) was prepared by adding 60 µL of dye into 6 mL antiserum for a final dilution of 1:100. The wells of the provided 96-well plate were washed five times with 300 µL of Wash Buffer (1×). Plate layout was followed as suggested by ELISA kit. 100 µL Immunoassay Buffer C (1×) was added to Non-Specific Binding (NSB) wells and 50 µL was added to Maximum Binding (B$_o$) wells. 50 µL of ELISA Standards were added to the respective wells. 50 µL samples were added to appropriate wells and 50 µL tracer was added to all wells except Total Activity (TA) and Blank (Blk) wells. 50 µL Antiserum was added to all wells except TA, NSB, and Blk wells. 96-well plate was covered and placed in BioTek SYNERGY H1 plate reader for 2 hours at room temperature with continuous orbital shaking. Wells were emptied and washed with 300 µL Wash Buffer (1×). 175 µL TMB Substrate Solution (#400074) was added to all wells and 5 µL tracer was added to TA wells. Plate was covered and placed in BioTek SYNERGY H1 plate reader for 30 minutes at room temperature with continuous orbital shaking. 75 µL of HRP Stop Solution (#10011355) was added to all wells. Plate was read in BioTek SYNERGY H1 plate reader at wavelength of 450 nm.

## Western blotting for type I fimbriae

Overnight cultures of hvKp NTUH-K2044 were synchronized for two hours to an $OD_{600}$ between 0.4 and 0.6. Synchronized cultures were treated with 7.5 µmol L$^{-1}$ bac7 for 30 minutes in a water bath at 37°C. Treatment with sterile 1 × PBS was used as a negative control (e.g., 0 µmol L$^{-1}$ bac7). Cultures were centrifuged, washed 1x with sterile 1 × PBS, and resuspended in 50µl of PBS. 6 × SDS and 1M HCL were added to samples that were then boiled at 95°C for 5 minutes. 1M NaOH was added to samples after cooling. After treatment, samples were run on SDS-PAGE gels. Gels were transferred to a 0.2 µm PVDF membrane using semi-dry transfer system. Transferred membranes were blocked overnight in 2% bovine serum albumin (BSA) and 3% milk in PBS. Membranes were washed (0.1% PBST) then stained with either 1:2,000 1°Rabbit anti-whole type I pili and 1:2,000 Rpo alpha subunit isolated from *E. coli* (loading control) for 2 hours rocking at room temperature. After 2 hours, membranes were washed and treated with 1:10,000 Goat anti-rabbit IgG HRP or 1:5,000 Sheep anti-mouse for 2 hours rocking at room temperature. Following

treatment with 2° antibodies, membranes were washed and exposed with ECL (Bio-Rad) before imaging. Blots were analyzed using ImageJ software and GraphPad Prism v10. Mouse anti-RNA Polymerase (Biolegend cat. 663104) was used for RNA-polymerase control staining. Rabbit anti-type 1 pilus antibody was produced by immunizing soluble pilus protein in rabbits by New England Peptide (now Biosynth, Louisville, KY). Whole pili were purified using the *E. coli* strain AAEC185 with the plasmid pBAD18-Cm-*fimA-H*, containing the pilus operon from *K. pneumoniae* TOP52. Cultures were grown overnight in LB at 37°C with 0.2% arabinose. Pili were purified by heating to 65°C, shearing via vortex, and salt precipitation with $MgCl_2$ as previously described [86]. The soluble pilus fraction was used for vaccination of rabbits and antibody preparation.

## Membrane depolarization assay

Bacterial cultures were grown overnight and synchronized as described in the bacterial killing assays. Following 3-hour synchronization, samples were washed three times with 15 mL Buffer A (5 mM HEPES, 5.0% glucose) and resuspended in 10 mL Buffer A. The resuspended cells were standardized to $OD_{600}$ 0.1 in 25 mL Buffer A with 100 mM KCl and incubated in shaking incubator (150 rpm) at 30°C for 15 minutes. Following the brief incubation, 2 µmol $L^{-1}$ $DiSC_3(5)$ (3,3'-Dipropylthiadicarbocyanine Iodide) (ThermoFisher) was added to the cell suspension and was pipetted into a 96-well black-walled clear bottom plate (Corning). Fluorescence measurements (excitation 622 nm, emission 670 nm) were recorded every 2 minutes at 37°C using BioTek SYNERGY H1 plate reader to detect fluorophore quenching. Following the initial 30-minute quenching, 2-fold bac7 dilutions using buffer A were prepared alongside positive control peptide cecropin A (Anaspec) and negative control ertapenem (Fisher Scientific) in triplicate as described in the minimal inhibitory concentration assay and 50 µL of the peptide dilutions were transferred to the black-walled clear bottom 96-well plate containing the cell suspension with quenched $DiSC_3$ dye. Fluorescence measurements (excitation 622 nm, emission 670 nm) were recorded every 2 minutes at 37°C for an additional 30 minutes. The fluorescence readouts were graphed with error reported as ±SEM. When comparing the correlation between membrane depolarization and mucoidy or colistin resistance, simple linear regression was used with Pearsons's correlation analysis to determine if there was a significant connection between the two variables.

## Single cell fluorescence microscopy imaging

Cells were prepared and imaged as previously described [87,88]. Briefly, cultures were streaked out and grown overnight at 37°C. Cultures were then standardized to an $OD_{600}$ of 0.1 in 2 mL in MHB and FITC-bac7 was added. Samples were incubated at 37°C shaking at 230 rpm for 30 minutes. 1 mL of culture was spun down and resuspended in 100 µl of supernatant. 5 µL of culture was spotted onto a glass bottom dish (Mattek, P35G-1.5-14-C) and cells were then stained with 1 µg $mL^{-1}$ FM 4–64 fluorescent dye (MilliporeSigma, 574799–5MG) to visualize the membrane and covered with an agarose pad (a process that takes 15 minutes which is included in the incubation times). Images were captured on a DeltaVision Core microscope system (Leica Microsystems) equipped with a Photometrics CoolSNAP HQ2 camera and an environmental chamber. Seventeen planes were acquired every 200 nm and the data was deconvolved using SoftWoRx software. Images were created using Fiji [89].

## Transmission electron microscopy

***Negative stain for membrane damage assessment.*** *K. pneumoniae* NTUH-K2044 was grown overnight and diluted to an $OD_{600}$ 0.1 in Mueller Hinton Broth with 7.5 µmol $L^{-1}$ bac7 and incubated for 15 minutes at 37°C. After peptide treatment for 15 minutes the media was removed, and the cells were resuspended in an equal volume of 0.2 × PBS so that the salts did not interfere with imaging. Formvar-carbon coated grids (EMS) were glow discharged for one minute before adding 5 µL of sample and the sample was allowed to sit for 2 minutes. Grids were then blotted, washed with a drop of water, blotted again, and then stained with 5 µL of 1% phosphotungstic acid (PTA) (pH 7) for 1 minute. The stained samples were blotted and allowed to air dry before imaging using a Tecnai Spirit TEM at 80kV. Images used for the figures

were taken at 26.5kx and 43.5kx magnification. The microscopy imaging was performed at the University of Texas Center for Biomedical Research Support (CBRS) core.

**Fimbriae analysis.** Strains were cultured as described above for Western blotting. NTUH-K2044 was grown overnight and synchronized to an $OD_{600}$ between 0.4-0.6. Cultures were washed with 1×PBS and resuspended in 1×PBS. Samples were then prepped for negative-stain EM by fixing in 1% glutaraldehyde for 10 minutes. The bacterial suspension was allowed to absorb onto freshly glow discharged formvar/carbon-coated copper grids (200 mesh, Ted Pella Inc., Redding, CA) for 10 minutes. Grids were then washed two times in dH2O and stained with 1% aqueous uranyl acetate (Ted Pella Inc.) for 1 minute. Excess liquid was gently wicked off and grids were allowed to air dry. Samples were viewed on a JEOL 1200EX transmission electron microscope (JEOL USA, Peabody, MA) equipped with an AMT 8-megapixel digital camera (Advanced Microscopy Techniques, Woburn, MA). Percent piliation was determined by manually counting at least 50 cells/sample and dividing the number of cells with extruding pili by the total number of cells counted.

## Inner membrane permeability assay

Bacterial cultures were grown overnight and synchronized as described in the bacterial killing assays. Following synchronization, the bacteria were centrifuged and washed with 25 mL of 100 mM sodium phosphate buffer. Centrifugation was repeated and bacteria was resuspended in 5 mL 100 mM sodium phosphate buffer and $OD_{600}$ reading was recorded. Bac7 was diluted alongside positive control peptide cecropin A and negative control ertapenem using 2-fold dilutions in 100 mM sodium phosphate buffer in a 96-well black-walled clear bottom plate (Corning). Culture was standardized to $OD_{600}$ 0.1 in 100 mM sodium phosphate buffer with 1.5 mM ONPG (o-nitrophenyl-β-D-galactopyranoside) was added to all wells. Fluorescence measurements (excitation 622 nm, emission 670 nm) were recorded every 2 minutes for 45 minutes using a BioTek SYNERGY H1 plate reader. The fluorescence readouts were graphed with error reported as ±SEM.

## Intracellular pH assessment

Bacterial cultures were grown overnight and synchronized as described in the bacterial killing assays. Following synchronization, 50 mL culture was centrifuged at 15,000×g for 10 minutes at room temperature. Supernatant was removed and bacterial pellet was resuspended in 10 mL 5 mM HEPES buffer with 5 mM glucose. Centrifugation was repeated with the same parameters above. Supernatant was removed and pellet was resuspended in 10 mL 5 mM HEPES buffer with 5 mM glucose. $OD_{600}$ reading was recorded, and culture was standardized to $OD_{600}$ 0.1 in 5 mM HEPES buffer with 5 mM glucose. 2-fold serial dilutions of drug were performed in 96-well black-walled clear bottom plate (Corning). 50 μL of standardized culture was added to wells of 96-well black-walled clear bottom plate (Corning) containing drugs and was incubated in 37°C static incubator for 30 minutes. 12 μmol $L^{-1}$ pHrodo red fluorescent dye (Invitrogen) was prepared in 5 mM HEPES buffer with 5 mM glucose. 100 μL of 12 μmol $L^{-1}$ pHrodo red fluorescent dye (Invitrogen) was added to wells of 96-well black-walled clear bottom plate (Corning) containing bac7, chloramphenicol, and polymyxin B and was incubated for 30 minutes at room temperature to allow for uptake of dye. Absorbance was read at 533 nm fluorescence reading using a BioTek SYNERGY H1 plate reader. The fluorescence readouts for n=3 replicates were graphed with error reported as ±SEM.

## Flow cytometry with FITC-bac7

To understand the heterogeneity of peptide uptake with a large population of cells we used flow cytometry following treatment with 1.9 μmol $L^{-1}$ FITC-bac7 for 30 minutes. Following treatment, the cells were centrifuged to remove remaining peptide and resuspended in phosphate buffered saline with 200 μmol $L^{-1}$ *Bac*Light Red Bacterial Stain (Invitrogen) and incubated in the dark at room temperature for 15 minutes. A CytoFLEX flow cytometer was used to analyze the samples using FITC laser and APC red fluorescent laser. The no peptide control was used to gate for no green fluorescence before analyzing the samples with FITC-bac7 treatment. FlowJo software was used to generate graphs used for the figures

PLOS Pathogens

shown. The samples were run in triplicate and the total count of FITC+/Red+ and FITC-/Red+ for all three samples were graphed with error shown as ±SEM.

## Bacterial killing assays

**Minimal inhibitory concentration assays.** Overnight cultures were grown in LB media at 37°C in shaking incubator (220 rpm) for 24 hours. Overnight cultures were synchronized 1:100 in LB media at 37°C in shaking incubator (220 rpm) for 3 hours and standardized to an $OD_{600}$ 0.002 in MHB1 media. Bac7 stock concentration (10 mg mL$^{-1}$) was diluted in BSA buffer (BSA and 0.1% acetic acid) to 32 μg mL$^{-1}$ in triplicate and diluted 2-fold in 96-well plate using BSA. Following drug dilution, 50 μL standardized culture was added to all wells containing 50 μL of the drug dilution. The 96-well plate was sealed using Parafilm and placed in 37°C static incubator for 24 hours. 96-well plate was read at $OD_{600}$ after 24 hours. For minimum inhibitory concentration assays testing clinical antibiotics, the methods detailed above were utilized with minor modifications. Overnight cultures were synchronized 1:100 in LB media at 37°C in shaking incubator (220 rpm) for 3 hours and standardized to an $OD_{600}$ 0.002 in Mueller Hinton Broth 2 (MHB2) media. Clinical antibiotics were diluted in MHB2 media to 128 μg mL$^{-1}$ and then diluted 2-fold in MHB2 media in a 96-well plate. 50 μL standardized cultures were added to each well containing 50 μL diluted drugs. The 96-well plate was sealed using Parafilm and placed in 37°C static incubator for 24 hours. 96-well plate was read at $OD_{600}$ after 24 hours.

**Time-kill assay.** Overnight cultures were grown in LB media at 37°C in shaking incubator (220 rpm) for 24 hours and synchronized 1:100 in LB media at 37°C in shaking incubator (220 rpm) for 3 hours and standardized to $OD_{600}$ 0.01 in MHB1 media. 2-fold dilutions of bac7 peptide were performed in MHB1 media to achieve 0.5×, 1×, and 4× MIC concentrations, respectively. At each timepoint, 100 μL from culture tubes was transferred to 96-well plate and serially diluted 10-fold in 1× PBS. 5 μL serial dilutions were spot plated onto LB agar plate and placed in 37°C static incubator for 24 hours. Bacterial growth was assessed after 24 hours by enumeration of CFU mL$^{-1}$. All time-kill assays were performed in biological triplicate with error reported as ±SEM.

## Biofilm viability assay

Clinical antibiotic MBEC assays were performed by growing overnight cultures in LB media at 37°C in shaking incubator (220 rpm) for 24 hours. Overnight cultures were standardized to $OD_{600}$ 0.5 (9.75 × 10$^9$) in biofilm media and 200 μL were pipetted into CBD (Innovotech) 96-well plate with peg lid and placed in 37°C static incubator for 24 hours. Chloramphenicol, ertapenem, and gentamicin were diluted from 64 μg mL$^{-1}$ to 0.5 μg mL$^{-1}$ using 2-fold dilutions in new 96-well plate using MHB2 media. Peg lid from CBD device was transferred to 96-well drug dilution plate, sealed with Parafilm, and placed in 37°C static incubator for 24 hours. Following incubation with the antibiotics, the peg lid was placed into 96-well plate containing 200 μL 0.1% Crystal Violet (CV) for 15 minutes. The peg lid was removed and dried in fume hood for 24 hours. De-staining was completed by solubilizing the CV stain on the peg lid in 30% acetic acid for 15 minutes and the solubilized stain was read at $OD_{550}$ using a BioTek SYNERGY H1 plate reader.

## Confocal microscopy z-stack imaging

**Biofilm growth and treatment.** Overnight cultures were grown in LB media in shaking incubator (220 rpm) at 37°C for 24 hours. Overnight cultures were standardized to $OD_{600}$ 0.5 in biofilm media and pipetted into #1.5 14 mm glass diameter Matsunami dishes (VWR). Matsunami dishes were sealed with Parafilm and placed in static incubator at 37°C for 24 hours. After 24 hours, supernatant was removed from the biofilms and replaced with MHB1 media with 15 μmol L$^{-1}$ bac7 or water only for no treatment control samples. Treatments performed for FITC- bac7 were performed as described above.

**Staining.** Post-24-hour treatment, supernatant of biofilms was removed, samples were washed with 1 mL 1× PBS and stained for imaging. To visualize the bacterial cells, the biofilms were stained with 5 μmol L$^{-1}$ SYTO 9 green fluorescent stain in 1× PBS using a rocker at medium speed for 1 hour. SYTO 9 was removed, samples were washed with 1× PBS,

and 50 µg mL⁻¹ calcofluor white dye in molecular-grade water was added and rocked at medium speed for 5 minutes to stain the polysaccharide matrix. Calcofluor white dye was removed, and samples were washed with 1 × PBS before confocal imaging. When visualizing biofilms treated with FITC-bac7, post-staining with calcofluor white stain was performed as described above. BactoView (red) cell stain was used in place of SYTO 9 at 5 µmol L⁻¹ in 1 × PBS with the same staining procedure as described above for SYTO 9.

**Imaging.** Biofilms were imaged using confocal z-stack imaging on a Zeiss LSM 710 confocal microscope with a 63 × oil immersion lens. Lens oil was applied to the microscope lens, and sample dish (Matsunami Glass dish) was placed onto the microscope lens holder. Laser channels 488 and 543 nm laser channels for imaging of calcofluor white and SYTO 9 staining, respectively. When using BactoView laser channels 588 and 543 nm were used to visualize the red cell fluorescence and calcofluor white polysaccharide stain, respectively. Images were taken using confocal z-stacks (0–60 µm) and 3D renderings of the z-stack images were generated using ZEN microscope software.

**BiofilmQ analysis.** Confocal z-stack imaging files were processed with BiofilmQ software package using MATLAB_ R2024b. Biofilm segmentation was done using a cube size of 1.32 µm resulting in 25 bins for MRSN 564304 and 15 bins for MRSN 1912. Fluorescence intensity as the bins increase in distance from the surface (dz (µm)) is measured to assess mean fluorescence through the layers of the biofilm. 4D XYZC plots were generated separately for red fluorescence and green fluorescence with the intensity of the respective signals displayed as heat map with the legend for each shown to the right.

## Skin abscess mouse model

Bacterial cultures were grown in tryptic soy broth (TSB) media until $OD_{600}$ 0.5 was achieved. Cells were washed twice with 1 × PBS and resuspended in 1 × PBS to reach a final concentration of $5 × 10^5$ CFU mL⁻¹. Six-week-old female CD-1 mice were anesthetized with isoflurane and subjected to a 1-cm-long superficial linear skin abrasion on their backs. 20 µL bacterial load resuspended in 1 × PBS was inoculated over the abraded area. Polymyxin B control and bac7 were diluted in water to 10 × the MIC and applied to the infected region 5 hours post-infection. The animals were euthanized 2 and 4 days after infection, skin samples and organs were collected and homogenized to enumerate the bacterial burden. Scarified skin areas and organ systems were excised and homogenized using a bead beater (25 Hz) for 20 minutes and serially diluted using 10-fold dilutions in 1 × PBS for CFU quantification. Six mice per group (n = 6) were used for the experimental groups and statistical significance was determined using one-way ANOVA with non-parametric Kruskal-Wallis test applied. p-values are shown for each of the groups, with all groups compared to the untreated control group. Mice were single-housed to avoid cross-contamination and maintained under a 12-hour light/dark cycle at 22°C with controlled humidity at 50%.

## Supporting information

**S1 Fig. KEGG enrichment for pathways and compartments.** The figure shows the total pathways effected by bac7 (1–35) treatment. Ribosome pathway is upregulated with bac7 treatment compared to the control no treatment sample and all other pathways shown enriched are downregulated with bac7 treatment compared to the control no treatment sample.
(TIF)

**S2 Fig. KEGG gene ontology enrichment.** The figures show the gene ontology of the changes associated with bac7 (1–35) treatment when considering subcellular localization (A) cellular component (B), molecular function (C), and biological processes (D).
(TIF)

**S3 Fig. STRING protein-protein interaction network.** The figure shows the protein-protein interactions between the gene expression changes from our RNA sequencing. The circles surround the genes involved in protein translation (pink circle) and oxidative phosphorylation (blue circle), respectively.
(TIF)

**S4 Fig. RT-qPCR of c-di-GMP regulated genes and Fimbriae analysis Immunoblot assays.** Panel A shows ATP at 30 minutes, 3 hours, and 6 hours with bac7 treatment compared to chloramphenicol and polymyxin B ribosomal binding and lytic peptide controls, respectively. Panel B shows RT-qPCR of *mgtC* 30 minutes following treatment with bac7 and controls. Panel C shows c-di-GMP quantification following treatment with bac7 for 30 minutes. ATP and c-di-GMP quantification was performed next to the respective ribosomal and lytic controls chloramphenicol (CHL) at 396 µmol L$^{-1}$, polymyxin B (PolyB) at 25 µmol L$^{-1}$. One-way ANOVA was used to determine significance for B and C with Dunnett's correction for multiple comparisons with adjusted p-values shown (asterisks indicate p-values **** $< 0.0001$, ** $< 0.01$, and ns $> 0.1$) and error shown reported as ±SEM. Panel D shows the Immunoblot analysis of *K. pneumoniae* NTUH K2044 with increasing concentrations of bac7 (1 - 7.5 µmol L$^{-1}$).
(TIF)

**S5 Fig. Permeability assay shows no leakage in cKp or hvKp when treated with bac7 (1–35).** Panel A shows a β-galactosidase leakage assay using o-nitrophenyl-β-d-galactopyranoside (ONPG) in cKp control MKP103 after treatment with bac7 (1–35), positive control cecropin A, and negative control ertapenem. Panel B displays ONPG assay in hvKp NTUH K2044 after treatment with bac7 (1–35), cecropin A, and ertapenem. Normalized fluorescence intensity values were achieved from subtracting blank 96-well plate intensity values by sample values. Panel C shows intracellular pH changes with treatment of chloramphenicol, bac7 (1–35), and polymyxin B. The fluorescence readouts were graphed with error reported as ±SEM.
(TIF)

**S6 Fig. Membrane depolarization of control hvKp and colistin resistant cKp.** The Figure shows the kinetic depolarization of control strains hvKp NTUH K2044 and cKp MKP103 with bac7 (1–35) (A and B), positive control lytic peptide cecropin A (C and D), and negative control non-lytic ertapenem (E and F). Cationic hydrophobic DiSC$_3$(5) (3,3'-Dipropylthiadicarbocyanine Iodide) dye was used. Readouts were measured in fluorescence intensity and normalized fluorescence intensity values were achieved by subtracting blank wells in 96-well plate from sample wells. Errors were reported as ±SEM.
(TIF)

**S7 Fig. Single cell fluorescence microscopy images show membrane and DNA staining along with progression of FITC-labeled bac7 (1–35) cytosol entry.** HvKp control strains NTUH K2044 (K1), KPPR1S (K2), and its capsular mutant Δ*wcaJ* were treated with 0 and 7.5 µmol L$^{-1}$ and incubated for 15 and 45 minutes (A). Cell membranes were stained with FM4–64 red dye and DAPI blue, fluorescent dye was used to stain the cellular DNA. B shows hvKp control strains NTUH K2044 (K1), KPPR1S (K2), and its capsular mutant Δ*wcaJ* treated with 1.9 FITC-labeled bac7 (1–35) for 15 and 45 minutes. Membrane and DNA staining described above were utilized.
(TIF)

**S8 Fig. Bac7 (1–35) treatment reveals differential membrane depolarization in high biofilm forming isolates.** The Figure shows the kinetic depolarization of control strains hvKp NTUH K2044 and cKp MKP103 as well as the highest biofilm-forming isolates in the MRSN diversity panel after treatment with bac7 (1–35) for 30 minutes. Cationic hydrophobic DiSC$_3$(5) (3,3'-Dipropylthiadicarbocyanine Iodide) dye was used. Readouts were measured in fluorescence intensity and normalized fluorescence intensity values were achieved by subtracting blank wells in 96-well plate from sample wells. Errors were reported as ±SEM.
(TIF)

**S9 Fig. Cecropin A treatment reveals sustained membrane depolarization in high biofilm forming isolates.** Kinetic depolarization of hvKp (NTUH K2044) and cKp (MKP103) control strains as well as the highest biofilm-forming isolates in the MRSN diversity panel are shown after treatment with cecropin A for 30 minutes. $DiSC_3(5)$ (3,3'-Dipropylthiadicarbocyanine Iodide) cationic hydrophobic dye was used. Readouts were measured in fluorescence intensity and normalized fluorescence intensity values were achieved by subtracting blank wells in 96-well plate from sample wells. Errors were reported as ±SEM.
(TIF)

**S10 Fig. Treatment with ertapenem displays the absence of membrane depolarization in both cKp and hvKp strains.** This Fig shows the depolarization of both cKp (MKP103) and hvKp (NTUH K2044) control strains and MRSN isolates with the highest biofilm formation abilities after 30-minute treatment with ertapenem. The cationic hydrophobic $DiSC_3$ (3,3'-dipropylthiadicarbocyanine Iodide) dye was utilized. Normalized fluorescence intensity values were achieved by subtracting blank wells of 96-well plate with sample wells. Errors were reported as ±SEM.
(TIF)

**S11 Fig. Variable membrane depolarization is shown in clinical isolates with increased %mucoviscosity.** Figure shows membrane depolarization intensities after treatment with bac7 (1–35) of cKp control MKP103 and hvKp control NTUH K2044 as well as clinical isolates from the MRSN diversity panel that displayed increased %mucoviscosity. $DiSC_3$ cationic dye was used. Readouts were measured with fluorescence intensity and were normalized by subtracting fluorescence intensity values of blank wells in 96-well plate with sample wells. Errors were reported as ±SEM.
(TIF)

**S12 Fig. Elevated fluorescence intensities are shown in clinical isolates with increased %mucoviscosity after treatment with cecropin A.** The Figure shows membrane depolarization kinetic graphs with MRSN clinical isolates treated with cecropin A that displayed the highest %mucoviscosity values in the diversity panel. cKp (MKP103) and hvKp (NTUH K2044) control strains are shown for reference. $DiSC_3$ dye was utilized. Normalized fluorescence intensity values were achieved by subtracting blank wells in 96-well plate from sample wells. Errors were reported as ±SEM.
(TIF)

**S13 Fig. HvKp and cKp isolates result in no depolarization after treatment with ertapenem.** The Fig shows membrane depolarization after treatment with ertapenem in both hvKp (NTUH K2044) and cKp (MKP103) control strains as well as MRSN clinical isolates that display the highest mucoviscosity values in the MRSN diversity panel. Assays were performed with $DiSC_3$ cationic dye. Normalized fluorescence intensity values were achieved by subtracting blank wells in 96-well plate from sample wells. Errors were reported as ±SEM.
(TIF)

**S14 Fig. Depolarization with positive control cecropin A and negative control ertapenem.** Membrane depolarization of MRSN isolates after 10 minutes treatment with 0.25 µmol $L^{-1}$ cecropin A (A) or 2.1 µmol $L^{-1}$ ertapenem (B). Normalized fluorescence intensity values were obtained by subtracting fluorescence intensities from the blank wells in 96-well plate from sample wells. Errors were reported as ±SEM.
(TIF)

**S15 Fig. Scatter plot correlation between membrane depolarization and %mucoviscosity and colistin resistance.** The figures show membrane depolarization of the MRSN isolates on the Y-axis and percent mucoviscosity or colistin resistance on the X-axis obtained from these isolates in our previous studies [15,66]. Simple linear regression was used to determine the slope with the 95% confidence interval and p-value of the slope shown. Correlation between the variables

was determined using Pearson correlation analysis with R-squared and p-values shown. Errors for membrane depolarization were reported as ±SEM.
(TIF)

**S16 Fig. Time kill assays show bacteriostatic effects after treatment with bac7 (1–35) in clinical *K. pneumoniae* isolates.** Time kill assay showing after treatment of clinical isolates MRSN 365679 (A) and 1912 (B) with bac7 (1–35) up to 24 hours. Samples were treated with 0, 0.5, 1 and 4X the minimum inhibitory concentrations (MICs), respectively with results measured in CFU mL$^{-1}$. Errors were reported as ±SEM.
(TIF)

**S17 Fig. Bac7 displays differential biofilm disruption with MRSN clinical biofilms.** The figure shows fold change MIC/MBEC$_{90}$ for common antibiotics (A) and bac7 (B) for MRSN isolates and hvKp NTUH-K2044. C-E show z-stack confocal microscopy images of untreated and treated biofilms stained with SYTO 9 (green cells) nucleic acid stain and calcofluor white dye (blue matrix polysaccharides). MRSN 1912 and 16008 biofilms treated with 15 µmol L$^{-1}$ bac7 are shown in C and D. MRSN 731029 and 564304 (E and F) are shown untreated, and treated with 15 µmol L$^{-1}$ bac7, 198 µmol L$^{-1}$ chloramphenicol (CHL). or a combination of bac7 and chloramphenicol. All MICs and MBEC$_{90}$s were identified using biological triplicate and n = 3 biofilms were imaged for confocal experiments with representative images shown.
(TIF)

**S18 Fig. ATP quantification of 564304 biofilms at 30 minutes and 3 hours treatment with bac7.** ATP quantification of MRSN 564304 biofilm embedded cells using relative luminescence units (RLU) normalized to the respective OD$_{600}$ (RLU/OD$_{600}$) using a CellTiter-Glo kit with no treatment and 15 µmol L$^{-1}$ bac7 for 0.5 and 3 hours (A). Biofilm OD$_{600}$ MRSN 564304 for each timepoint used to normalize ATP RLU (B). One-way ANOVA was used to determine significance compared to the no treatment groups for A and B with Dunnett's correction for multiple comparisons and adjusted p-values shown (asterisks indicate p-values **** < 0.0001, ** < 0.01, and ns > 0.1) and error shown reported as ±SEM.
(TIF)

**S19 Fig. Membrane depolarization with transposon mutants.** Figure shows membrane depolarization intensities after treatment with bac7 (1–35) and cecropin A positive control of MKP103 Δ*mgtC* and Δ*sbmA* transposon mutants. DiSC$_3$(5) (3,3'-Dipropylthiadicarbocyanine Iodide) cationic dye was used. Readouts were measured with fluorescence intensity and were normalized by subtracting fluorescence intensity values of blank wells in 96-well plate with sample wells. Errors were reported as ±SEM. E and F Fig show intracellular c-di-GMP levels following treatment for 30 minutes of MKP103 WT and Δ*mgtC* mutant, respectively with bac7 8 µmol L$^{-1}$, tetracycline and cecropin A controls using a cyclic di-GMP ELISA kit (Fig 1E). Polymyxin B resistant MKP103 and chloramphenicol transposon insert used for mutant creation lead to variable controls compared to NTUH K2044.
(TIF)

**S20 Fig. *In vivo* skin abscess bacterial recovery on day 2.** The figure shows the CFU mL$^{-1}$ of the skin samples and bacteria disseminated to the kidney, spleen, and liver. Data was graphed as an interleaved scatter plot to show the mean values. Significance was determined using one-way ANOVA and p-values were obtained by comparing untreated groups to the treated groups.
(TIF)

**S1 Table. Kegg pathway analysis results.**
(DOCX)

**S2 Table. Lab strains used in this study.**
(DOCX)

**S3 Table. Clinical isolates used in this study.**
(DOCX)

**S4 Table. List of primer pairs used for RT- qPCR.**
(DOCX)

**S1 Data. RNA sequencing data.**
(XLSM)

**S2 Data. Genes uploaded to KEGG pathway analysis.**
(XLSX)

## Author contributions

**Conceptualization:** Renee M. Fleeman.

**Data curation:** Robert L Beckman IV, Berta Martinez, Flor Z Santiago, Gabriela N Echeverria, Marcelo D.T. Torres, Logan Suits, Shantal Garcia, Paeton L Wantuch, Cesar de la Fuente-Nunez, Prahathees Eswara, David A Rosen, Renee M. Fleeman.

**Formal analysis:** Berta Martinez, Bruno V. Pinheiro, Marcelo D.T. Torres, Shantal Garcia, Paeton L Wantuch, Cesar de la Fuente-Nunez, Prahathees Eswara, David A Rosen, Renee M. Fleeman.

**Funding acquisition:** Cesar de la Fuente-Nunez, Renee M. Fleeman.

**Investigation:** Renee M. Fleeman.

**Methodology:** David A Rosen, Renee M. Fleeman.

**Project administration:** Renee M. Fleeman.

**Resources:** Renee M. Fleeman.

**Software:** Renee M. Fleeman.

**Supervision:** Cesar de la Fuente-Nunez, Prahathees Eswara, David A Rosen, Renee M. Fleeman.

**Validation:** Renee M. Fleeman.

**Visualization:** Renee M. Fleeman.

**Writing – original draft:** Robert L Beckman IV, Marcelo D.T. Torres, Cesar de la Fuente-Nunez, David A Rosen, Renee M. Fleeman.

**Writing – review & editing:** Robert L Beckman IV, Marcelo D.T. Torres, Cesar de la Fuente-Nunez, Prahathees Eswara, David A Rosen, Renee M. Fleeman.

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
