## [Decision Letter · Decision Letter 0]

11 Sep 2025

Molecular response to the non-lytic peptide bac7 (1-35) triggers disruption of Klebsiella pneumoniae biofilm

PLOS Pathogens

Dear Dr. Fleeman,

Thank you for submitting your manuscript to PLOS Pathogens. Your manuscript was evaluated by members of the editorial board and two external referees. All were enthusiastic about the study (see below), but had substantial concerns, particularly about the presentation of the work. Therefore, we invite you to submit a revised version of the manuscript that addresses the points raised during the review process. We ask you to respond appropriately to all of the issues raised by both reviewers in your revision, which will entail considerable reorganization and rewriting of the manuscript as well as culling of some experimental data.

Please submit your revised manuscript within 60 days Nov 10 2025 11:59PM. If you will need more time than this to complete your revisions, please reply to this message or contact the journal office at plospathogens@plos.org. Please include the following items when submitting your revised manuscript:

We look forward to receiving your revised manuscript.

Kind regards,

R. Martin Roop II

Guest Editor

PLOS Pathogens

D. Scott Samuels

Section Editor

Editor-in-Chief

PLOS Pathogens

orcid.org/0000-0003-2946-9497

Editor-in-Chief

PLOS Pathogens

orcid.org/0000-0002-7699-2064

**Journal Requirements:**

At this stage, the following Authors/Authors require contributions: Robert L Beckman IV, Berta Victoria, Flor Z Santiago, Gabriela N Echeverria, Bruno Pinheiro, Marcelo D.T. Torres, Logan Suits, Shantal Garcia, Paeton L Wantuch, Cesar de la Fuente-Nunez, Prahathees Eswara, David A Rosen, and Renee M Fleeman. Please ensure that the full contributions of each author are acknowledged in the "Add/Edit/Remove Authors" section of our submission form.

- ® on pages: 32, 33, 36, 38, 39, 47, and 49.

Potential Copyright Issues:

- Figure 6. Please confirm whether you drew the images / clip-art within the figure panels by hand. If you did not draw the images, please provide (a) a link to the source of the images or icons and their license / terms of use; or (b) written permission from the copyright holder to publish the images or icons under our CC BY 4.0 license. Alternatively, you may replace the images with open source alternatives. See these open source resources you may use to replace images / clip-art:

5) Please ensure that the funders and grant numbers match between the Financial Disclosure field and the Funding Information tab in your submission form. Note that the funders must be provided in the same order in both places as well.

**Reviewers' Comments:**

Reviewer's Responses to Questions

**Part I - Summary**

Reviewer #1: In this manuscript by Beckman and colleagues, the authors seek to define the molecular mechanisms underpinning interactions between bac7 and K. pneumoniae biofilms. Using in vitro and in vivo approaches, the authors identify two mechanisms used by bac7 to elicit stress on K. pneumoniae. The authors support these findings through transcription analysis and elegant microscopy techniques. I particularly appreciated the thorough analysis of multiple K. pneumoniae strains and leveraging an in vivo model. My major concerns are text-based. The introduction was not broad enough to introduce the rationale of many subsequent experiments, and it was not until the Discussion that many aspects of bac7, biofilms, and ATP (for example) became clear. I also had difficulties with some terminology throughout, which is detailed in the Major Concerns section below. In summary, the authors have presented a thorough manuscript that would be improved with some alterations their descriptions of the data.

Reviewer #2: In this manuscript (PPATHOGENS-D-25-01948), the authors present data to begin to understand how the peptide bac7 disrupts biofilms and kills K. pneumoniae. Beginning with transcriptomics to determine changes in gene expression in response to bac7, they identified several pathways impacted and explored a couple of these with follow-up assays. They demonstrate that bac7 causes membrane depolarization but not lysis, and that it becomes internalized in a set of lab strains as well as with a cohort of clinical isolates. Use of mutants led to the identification of a transporter that partly responsible for uptake of bac7, and that MgtC protects from biofilm clearance by mitigating toxic ATP-levels. They end the study by testing the ability of bac7 to alleviate infection in a mouse model, finding that it reduced bacterial burden and prevented Klebsiella-mediated weight loss. Overall this is an important study as bac7 seems to be an effective treatment for Klebsiella biofilms and infections. While I encountered some confusion, I think, for the most part, the data are convincing and the conclusions sound.

There is a ton of data in this manuscript, and the text is a bit wordy. Personally, I think some editing and streamlining will improve the readability. In some cases, extra detail made it harder to read (minor comment #1), yet in others, I felt more detail would improve clarity (e.g. major comment #1). This comes with the caveat that a few of the assays here are outside my expertise, so this could be the reason for many of my comments below. As a side note, I would like to add my appreciation for figures that were large enough to easily evaluate!

**Part II – Major Issues: Key Experiments Required for Acceptance**

Reviewer #1: 1. Throughout the manuscript, the authors use the terms “biofilm dispersal” and “dissemination”. As someone familiar with the term dissemination but not as familiar with the idea of dispersal, I found this to be confusing. It seems that biofilm dispersal would be a form of dissemination since the bacteria would be breaking free of the matrix and therefore able to spread across sites. However, I think the authors use this term to imply the biofilms are disrupted – which means they can be more easily cleared by the immune system and then not disseminate. Can the authors clarify these terms somewhere in the text? Alternatively, they could use another term such as “biofilm disruption” but the authors should decide which is more appropriate given their expertise.

a. In a similar manner for terminology, Figure 6C shows that bac7 decreases initial site K. pneumoniae in the wound model but it does not necessarily show that bac7 decreases the process of dissemination. Dissemination is an active process in which bacteria migrate across host barriers to access secondary sites. In Figure 6C, it’s possible that bac7 decreased initial site CFU to a level in which dissemination was ineffective. When there are differences in initial site CFU, bacterial barcoding is required to understand differences in dissemination dynamics (PMID34636322). The authors should clarify they did not specifically measure dissemination in the present study, they measured colonization of primary and secondary sites.

2. The Introduction does not provide sufficient background for someone not in the biofilm field to understand the main text panels. A lot of this information is in the Discussion, but it would be helpful if it were moved to the Introduction section.

a. As someone familiar with K. pneumoniae pathogenesis, but not in the context of biofilms, I am not sure why ATP and cyclic-do-GMP levels are important for biofilms aside from being involved in general cell signaling.

b. In the Introduction, bac7 is suddenly introduced in Line 87. As someone unfamiliar with this peptide, the current set up does not sufficiently describe this. Is bac7 a neutrophil derived peptide? What does it seem to be doing in other bacterial species? Why would you hypothesize the role would be the same/different in K. pneumoniae?

c. In the Introduction, the paragraph in lines 68-79 describe various aspects of biofilms but these structures can widely vary between species. Can the authors clarify in the text when these references are specifically describing K. pneumoniae biofilms or features from other species? Also, can the authors provide a few sentences in this paragraph or the next on what is known about K. pneumoniae biofilms in general (what is the state of the field for this species)?

Reviewer #2: 1. There were numerous situations where it was not clear to me what aspect of the data lead to the stated interpretations. Some of these assays are new to me and my lack of familiarity meant I could not quickly understand how the authors reached a conclusion. I think a smidge more detail in the text would help. Think of this as the alt text that appears with images online. One example is with Fig. 2E. I didn’t get how it was determined the peptide had moved to the cytoplasm. I had to piece together info from the legend and opening the electronic figure because the green color in the intracellular space was not evident on my printout (yes, some of us still print!). I’d like to think that if I saw the green, it would have been immediately clear to me, but not all readers will understand the effect of merged images, and others are colorblind. A simple addition of something such as “green fluorescence in the cytosolic space indicated the peptide had translocated across the membrane” would help those of us less familiar with various assays. Similarly, in Fig. 2F, I can see what are likely the blebs, but what is the evidence of the capsular changes?

2. The number of DEG in response to bac7 exposure seems concerningly high. This is nearly half the genome! I think it would be worth addressing this, and perhaps ensuring the Log2FC and FDR cut-offs are consistent with similar studies (presentation of the p-values as -log10FDR is new to me and I didn’t do math to see if it is consistent with other reports I’ve seen).

3. Fig 1F-G, is it known if the RNAP alpha subunit is not feedback regulated? If rpoB and rpoC are impacted by bac7 treatment, then RNAP components may not be a suitable loading control. Minor: The legend for G could use more detail—how quantified, normalized, etc., and I suggest replacing “RNAP” in 1F with alpha subunit.

4. In Fig. 1H, it is really hard to see the fimbriae without zooming in on the electronic file. As this is not ideal, I suggest moving Fig. S4D to Fig. 1. This partly stems from my current question about the quantification of the westerns. The TEM are much more convincing of the decrease, but too hard to see in this main figure (even with the hi-res version).

5. Regarding the hypothesis that type 1 fimbriae are important for biofilms, is there evidence for this in Kleb? From what I recall, it has only been experimentally demonstrated for the type 3 fimbriae. If there are references supporting this, they should be included. I found it curious the authors chose to leave this as a hypothesis, rather than including assays with a fim mutant to test this. I know these mutants exist in KPPR1S and may be available in the MKP103 library. Inclusion of these mutants could bring the story to full circle with the mgtC element. Alternatively, a plasmid overexpressing a diguanylate cyclase that leads to increased Fim production might similarly work.

6. Regarding the enormous amount of data with the MRSN collection, I have to confess I was left unclear as to what ultimately was learned from this data, and this is in part due to my unfamiliarity with the assays. I think some streamlining of the text might help make it clearer to less informed audiences.

7. The section that begins on line 373, I was confused about the choice to use the sbmA mutant. There are a million transporters, so I was left wondering why this one? There is a reference mentioned in the discussion that this protein had been shown to transport bac7. This rationale should be included here in the Results.

8. Given that colistin resistance can arise from mutations within the PhoPQ system, I’m a bit concerned about the use of MKP103 in Figure 5, especially since this strain did not display the expected increase in ATP in response to bac7. I cannot remember what the mechanism of colistin resistance is in this particular strain, but whether it is dependent or independent of the PhoPQ system, this should probably be addressed.

9. I noted the choice of a one-way ANOVA to analyze the significance of the mouse data. I’m far from a stats expert, but as I understand, this is a parametric test, and mouse data is usually considered non-parametric. In addition, it is more common to report median values rather than means. I comment on this just to ask that the authors ensure an appropriate test was chosen.

**Part III – Minor Issues: Editorial and Data Presentation Modifications**

Reviewer #1: 1. In Figure 1, can the authors provide context for the concentration of bac7 used in these experiments? If so many genes are differentially regulated via RNAseq in the presence of 7.5 umol bac7, should a lower concentration have been used? Potentially, this could have illuminated specific stress pathways rather than a global stress regulator. The data from Figure 2A suggest that lower concentrations could be used to elicit similar membrane stress responses. In Figure 1H, can the authors highlight where on the pictures they are observing less fimbriae? In the current images, the thickness of the cells and surrounding area looks similar.

2. Figure 3A-B are difficult to read as the two time points are plotted together across multiple strains. Could the authors keep the most relevant timepoint in the main text and move the other timepoints to the supplement? For Figure 3, it is difficult for a reader to keep track of the HMV state of each strain in the text and then find it in the corresponding Main/Supplement figures. Could the authors consider a panel where maybe they plot HMV levels on one axis and the DiSC3 RFU on the other axis, and then perform a correlation between the two variables? That type of panel may be more effective in getting the point across that HMV is linked to membrane depolarization.

3. In Figure 6, why did the authors use polymyxin B and not chloramphenicol? Given the results in Figure 4, it would be really striking if bac7 and chloramphenicol worked well independently but eradicated infection when used together. I do not think this experiment is necessary for publication, but would be very compelling when used in conjunction with the evidence in Figure 4.

4. In a KPPR1 bacteremia TnSeq study, SbmA was identified as a fitness factor during bloodstream infections (PMID37463183). Can the authors comment on how SbmA may be contributing to virulence?

5. Can the authors clarify the dose of NTUH used in the mouse model? The text described it as CFU/mL but does not specific the mLs used, which would indicate the total CFU administered to the mice.

Reviewer #2: 1. I suggest adding all genes mentioned in the results to Table 1. I found the long list of LogFC & p-values to be distracting as I read, and this isn’t essential to understanding their relevance (can at least delete the p-value). If this info is in the table, it can be removed from the text. Also, this is probably Log2FC? That ‘2’ is pretty important.

2. In the paragraph beginning on line 171, there are several places where I got confused or felt more detail was needed. This is an important section for the subsequent experiments, and I think some editing will help clarify the logic.

--Line 175, the phrase “mgtC upstream hairpin loops” is a bit confusing. Do you mean hairpins within the 5’ UTR? I suggest rephrasing this to be clearer/more specific.

--Line 178, it is not immediately clear to my why reduced ATP synthase activity results in the depletion of c-di-GMP. Is there additional detail on how/why this happens? And is it really depleted, or just reduced?

--Line 181, “Repeating the bac7 treatment used for…” This caused me to back up and re-read a couple of times. Perhaps something like “using the same conditions as for…” would help?

3. Lines 228-229, I think this is referring to the fluorescence intensity following bac7 (or control) treatment rather than intensity “of bac7”? I initially read this as a control for background fluorescence of the compound, but then questioned what was being referred to here. Had to look at the supplemental figure to know what this meant.

4. Line 244, I think this should be referencing 2E rather than 2C? I noticed a few other minor typos and assume these will be caught in revision.

PLOS authors have the option to publish the peer review history of their article (what does this mean? ). If published, this will include your full peer review and any attached files.

**Do you want your identity to be public for this peer review?** For information about this choice, including consent withdrawal, please see our Privacy Policy .

Reviewer #1: No

Reviewer #2: No

**Figure resubmission:**

**Reproducibility:**



---

## [Decision Letter · Decision Letter 1]

14 Nov 2025

Dear Renee,

We are pleased to inform you that your manuscript 'Molecular response to the non-lytic peptide bac7 (1-35) triggers disruption of *Klebsiella pneumoniae* biofilm' has been provisionally accepted for publication in *PLOS Pathogens* .

Thank you again for supporting Open Access publishing; we are looking forward to publishing your work in *PLOS Pathogens* .

Sincerely,

Marty Roop

Guest Editor

*PLOS Pathogens*

D. Scott Samuels

Section Editor

*PLOS Pathogens*

Sumita Bhaduri-McIntosh

Editor-in-Chief

*PLOS Pathogens*

orcid.org/0000-0003-2946-9497

Michael Malim

Editor-in-Chief

*PLOS Pathogens*

orcid.org/0000-0002-7699-2064

Reviewer Comments (if any, and for reference):

Reviewer's Responses to Questions

**Part I - Summary**

Reviewer #1: The authors have fully addressed my concerns in this revised manuscript. I remain enthusiastic about this study and have no further comments.

Reviewer #2: Review of revised manuscript PPATHOGENS-D-25-01948-R1

I think the authors have satisfactorily addressed all of my concerns, and seem to have addressed those of Reviewer 1. The manuscript is much smoother and easier to read now. The inclusion of additional information and clarification has made it much more understandable—and more interesting! Just an FYI to the authors, I did spot a few very minor typos but neglected to note them. Might be on the lookout for these at the next steps.

**Part II – Major Issues: Key Experiments Required for Acceptance**

Reviewer #1: (No Response)

Reviewer #2: NA

**Part III – Minor Issues: Editorial and Data Presentation Modifications**

Reviewer #1: (No Response)

Reviewer #2: NA

PLOS authors have the option to publish the peer review history of their article (what does this mean? ). If published, this will include your full peer review and any attached files.

**Do you want your identity to be public for this peer review?** For information about this choice, including consent withdrawal, please see our Privacy Policy .

Reviewer #1: **Yes: ** Caitlyn L. Holmes, Ph.D.

Reviewer #2: No

---

## [Editor Report · Acceptance letter]

Dear Assistant Professor Fleeman,

We are delighted to inform you that your manuscript, " 

Molecular response to the non-lytic peptide bac7 (1-35) triggers disruption of Klebsiella pneumoniae biofilm," has been formally accepted for publication in PLOS Pathogens.

Best regards,

Sumita Bhaduri-McIntosh

Editor-in-Chief

PLOS Pathogens

orcid.org/0000-0003-2946-9497

Michael Malim

Editor-in-Chief

PLOS Pathogens

orcid.org/0000-0002-7699-2064